# A Discretization Framework for Robust Contextual Stochastic Optimization

**Rares Cristian, Georgia Perakis**
Operations Research Center
Massachusetts Institute of Technology, Cambridge, MA, USA
{raresc,georgiap}@mit.edu

## Abstract

We study contextual stochastic optimization problems. Optimization problems have uncertain parameters stemming from unknown, context-dependent, distributions. Due to the inherent uncertainty in these problems, one is often interested not only in minimizing expected cost, but also to be robust and protect against worst case scenarios. We propose a novel method that combines the learning stage with knowledge of the downstream optimization task. The method prescribes decisions which aim to maximize the likelihood that the cost is below a (user-controlled) threshold. The key idea is (1) to discretize the feasible region into subsets so that the uncertain objective function can be well approximated deterministically within each subset, and (2) devise a secondary optimization problem to prescribe decisions by integrating the individual approximations determined in step (1). We provide theoretical guarantees bounding the underlying regret of decisions proposed by our method. In addition, experimental results demonstrate that our approach is competitive in terms of average regret and yields more robust solutions than other methods proposed in the literature, including up to 20 times lower worst-case cost on a real-world electricity generation problem.

## 1 Introduction

In recent years, the field of machine learning (ML) has made remarkable strides in developing powerful algorithms that can automatically extract patterns and insights from data. While prediction is often the primary focus in many ML applications, the ultimate goal is to make optimal decisions based on these predictions. For example, one may predict the hourly electricity demand of a power plant for the next day. But more importantly, based off of this forecast, the operator must decide how much electricity to generate in order to minimize cost while staying within the operational constraints of the plant. Another key factor is the possible distribution of the uncertain parameters and being able to protect against worst-case scenarios. In the previous example, the decision-maker may have the goal to maximize the probability that their operational cost is below a certain threshold. Robustness is a crucial property in real-world decision-making since a single significantly poor decision may have separate damaging effects. For instance, it could damage a company's reputation and trust with its customers.

Traditionally, a predict-then-optimize approach has been used in practice: the learning stage is performed separately from the optimization task. First one trains a model to predict the uncertain parameters (such as the electricity demand), then independently solve the corresponding optimization problem. Recent work in *end-to-end* learning has focused on how to train a model with a loss function that is meant to explicitly approximate the true decision cost a prediction would produce. However these approaches only target minimizing the average cost, and do not, in general, take robustness into account. In this work, we propose a different paradigm for combining the learning and optimization tasks. In particular our paper makes the following contributions.

1) **Novel approach to contextual stochastic optimization problems that is robust and data driven:** We propose a novel data-driven method to tackle contextual stochastic optimization problems. The proposed method is directly applicable to any class of optimization problems,

including *linear, nonlinear* and *discrete optimization problems*. Furthermore, it gives rise to solutions that are *robust against uncertainty in the objective function* using a single user-defined parameter to control the degree of robustness. In contrast to more traditional robust optimization methods, our proposed approach does not rely on constructing uncertainty sets. It is data driven and does not need to make any assumptions about the structure of the data itself and its distribution.

2) **Analytical guarantees on the regret and the stability of the proposed approach:** We prove *analytical guarantees* on the *regret* of the proposed method. For instance, we show that the difference between the in-sample cost and the out-of-sample cost decreases on the order of $1/\sqrt{n}$, for $n$ datapoints. Furthermore, we prove the proposed method is *stable against noise*, showing that the decisions prescribed do not change significantly if the dataset is perturbed by noise.

3) **Computational experiments on a variety of applications:** We show with computational experiments that the proposed method is competitive in terms of the average error relative to existing approaches in the literature. In addition to testing our approach for linear optimization applications such as portfolio optimization using historical stock data, we also consider nonlinear optimization problem applications such as inventory allocation and electricity generation using real-world data. Finally, through these xperiments, we show *significant improvement in terms of robustness*. We obtain as much as 20 times lowers cost in the worst case when compared to other end-to-end learning methods and 5 times lower than other robust approaches.

## 2 RELATED WORK

Due to space limitations we will keep this section short.

**End-to-End Learning:**

Traditionally, the simplest way to learn the uncertain parameters is to do so independently of the optimization problem by minimizing a loss function such as mean-squared error between predictions and observed realizations. However, it has been shown that solving the predictive and decision-making problems independently, can produce significantly suboptimal decisions Cameron et al. (2021). As such, a large stream of the literature includes end-to-end methods where the goal is to propose predictions whose corresponding optimal decisions minimize the downstream task loss (e.g., the objective function). One of the earlier works related to end-to-end learning is Kao et al. (2009) which trains a model to minimize task loss of an unconstrained quadratic optimization problem. In general, the primary difficulty in end-to-end learning approaches is the differentiability of the constrained optimization task. Amos & Kolter (2017) extends the setting in Kao et al. (2009) to constrained quadratic optimization, computing the gradient by differentiating through the KKT system of equations at the optimal solution. Unfortunately, for linear optimization, the problem becomes more complex since the gradient of the output of a linear problem with respect to its objective coefficients is either zero everywhere or undefined. Wilder et al. (2019) addresses the issue by taking a similar approach to Amos & Kolter (2017) for linear optimization problems but also adding a quadratic regularization term to the objective function. Other approaches have focused on other methods of altering the loss function or the objective function to compute more useful gradients end-to-end. For instance, for linear optimization problems, Elmachtoub & Grigas (2022) constructs a surrogate loss function that is a convex and differentiable approximation of the objective function. Elmachtoub et al. (2020) takes this approach and proposes a method to train decision trees with this surrogate loss. Mandi & Guns (2020), Vlastelica et al. (2020), Berthet et al. (2020) take different approaches to address this issue. Kotary et al. (2021) and the references therein provide a general survey for end-to-end combinatorial learning problems.

The approach we propose is applicable directly to any class of optimization problems, while individual end-to-end methods are usually restricted to certain sub-classes of problems. Moreover, a major difference in the approach proposed in this paper, is that it is non-parametric and proposes decisions directly from data without requiring an intermediate forecast.

**Prescriptive Analytics and Robust Optimization:** To solve a stochastic optimization problem one can apply well-known methods such as contextual Sample Average Approximation (SAA) (see for example, Kleywegt et al. (2002)). The work of Bertsimas & Kallus (2020) extends SAA to take advantage of the contextual nature of the problem by using covariates and weighing samples in a

non-uniform way (unlike SAA) using ML methods such as $k$-nearest neighbors or decision trees. For instance, for an out-of-sample set of features $\boldsymbol{x}$, use the $k$-nearest observations in the training data to make a decision. Alternatively, Bertsimas & Koduri (2021) generates weights using global methods, not only by using data around a neighborhood of out-of-sample $\boldsymbol{x}$. Bertsimas et al. (2019a) extends the general methodology by introducing an optimal prescriptive tree framework to produce weights that are directly dependent on the optimization problem and minimize task loss. Furthermore, Kallus & Mao (2022) considers a similar framework using a random forest. Finally, Bertsimas & McCord (2019) applies these prescriptive ideas to the multi-period setting.

There has been significant work within the robust optimization literature over the years (see for example, the books by Ben-Tal et al. (2009), Bertsimas & den Hertog (2021) as well as the survey paper by Bertsimas et al. (2010) and the references within). In robust optimization, careful construction of the underlying uncertainty sets is required to ensure the models are not overly conservative. Various formulations have been proposed, starting for example, with Soyster (1973), Ben-Tal & Nemirovski (2000), and Bertsimas & Sim (2004). Nevertheless, uncertainty sets can be learnt as we gain information from data. Earlier papers uses estimates of the mean and standard deviation from the available data such as for example, Bertsimas et al. (2013) who takes a data-driven robust optimization view. Uncertainty sets could vary as a function of the features, as for example in Bertsimas & Van Parys (2021), Kannan et al. (2020) and Bertsimas et al. (2019b).

## 3 THE FRAMEWORK

In this section, we first formally describe the problem and the data-driven setting we study in this paper. Given a feasible region $\mathcal{P}$ and decision variables $\boldsymbol{w} \in \mathcal{P}$, the goal is to minimize an objective function $g_{\boldsymbol{\nu}}(\boldsymbol{w})$ parameterized by uncertain parameters $\boldsymbol{\nu}$. If we have exact knowledge of the realized uncertainty $\boldsymbol{\nu}$ values, the optimal decision could be determined through the following problem

$$w^*(\boldsymbol{\nu}) = \arg \min_{\boldsymbol{w} \in \mathcal{P}} g_{\boldsymbol{\nu}}(\boldsymbol{w}). \tag{1}$$

If the problem above does not a unique solution, we can instead assume that $w^*(\boldsymbol{\nu})$ is an oracle providing any optimal solution. For example, $g_{\boldsymbol{\nu}}(\boldsymbol{w})$ can correspond to a linear optimization problem objective, $g_{\boldsymbol{\nu}}(\boldsymbol{w}) = \boldsymbol{\nu}^T \boldsymbol{w}$ or a quadratic optimization problem objective, $g_{\boldsymbol{\nu}}(\boldsymbol{w}) = \boldsymbol{q}^T \boldsymbol{w} + \boldsymbol{w}^T \mathbf{Q} \boldsymbol{w}$, where $\boldsymbol{\nu} = (\boldsymbol{q}, \mathbf{Q})$ corresponds to the linear and quadratic objective coefficients respectively. In a shortest path example, the uncertainty parameters $\boldsymbol{\nu}$ would correspond to the unknown travel times along each edge while the decision $\boldsymbol{w}$ would be a vector determining which path to take. Naturally set $\mathcal{P}$ constrains $\boldsymbol{w}$ to properly satisfy the path related constraints. Finally, we formally define the notion of regret of a decision:

**Definition 1** *The regret $R_{\boldsymbol{\nu}}(\boldsymbol{w})$ of a decision $\boldsymbol{w}$ with respect to a parameterization $\boldsymbol{\nu}$ is given by the difference between its objective value and the optimal one corresponding to $\boldsymbol{\nu}$:*

$$R_{\boldsymbol{\nu}}(\boldsymbol{w}) = g_{\boldsymbol{\nu}}(\boldsymbol{w}) - g_{\boldsymbol{\nu}}(w^*(\boldsymbol{\nu})) \tag{2}$$

We make the following assumptions regarding the optimization problem:

**Assumption 3.1** *We assume that the maximum regret $R_{\boldsymbol{\nu}}(\boldsymbol{x}) = g_{\boldsymbol{\nu}}(\boldsymbol{w}) - g_{\boldsymbol{\nu}}(w^*(\boldsymbol{\nu}))$ is bounded and at most $M_1 > 0$ for any $\boldsymbol{\nu}, \boldsymbol{w}$.*

In the data-driven setting, we assume that the objective's uncertain parameters are distributed according to an unknown distribution $D_{\boldsymbol{x}}$ which depends on features $\boldsymbol{x}$. Given some vector of features $\boldsymbol{x}$, we need to compute decision $\hat{w}(\boldsymbol{x})$. Only afterwards can we observe the realization of $\boldsymbol{\nu}_{\boldsymbol{x}} \sim D_{\boldsymbol{x}}$ and incur a cost of $g_{\boldsymbol{\nu}_{\boldsymbol{x}}}(\hat{w}(\boldsymbol{x}))$. We take a data driven approach and do not assume we know the distribution $D_{\boldsymbol{x}}$ of the cost vector for any given feature vector $\boldsymbol{x}$. Rather we assume we are given $N$ data points $(\boldsymbol{x}^1, \boldsymbol{\nu}^1), \ldots, (\boldsymbol{x}^N, \boldsymbol{\nu}^N)$ consisting of observed covariates $\boldsymbol{x}^i$ and observed realizations $\boldsymbol{\nu}^i \sim D_{\boldsymbol{x}^i}$. One objective is to minimize the expected regret of the decision:

$$\min_{\boldsymbol{w} \in \mathcal{P}} \mathbb{E}_{\boldsymbol{\nu}_{\boldsymbol{x}} \sim D_{\boldsymbol{x}}} [R_{\boldsymbol{\nu}_{\boldsymbol{x}}}(\boldsymbol{w})] \tag{3}$$

while another would be to provide a solution that is more robust to uncertainty. One may wish to minimize the probability that the regret is above a certain threshold:

$$\min_{\boldsymbol{w} \in \mathcal{P}} \mathbb{P}_{\boldsymbol{\nu}_{\boldsymbol{x}} \sim D_{\boldsymbol{x}}} (R_{\boldsymbol{\nu}_{\boldsymbol{x}}}(\boldsymbol{w}) \geq \phi). \tag{4}$$

However, this formulation, even given perfect knowledge of the distribution of $w^*(\nu_x)$ is not tractable. For instance, for discrete $D_x$, it may be solved only by a mixed integer optimization problem (see Appendix E). We propose an approximation to this objective: to minimize the expected regret that is larger than $\phi$, which we denote as the *minimum violation* solution. If the regret of $w$ is below $\phi$ for a given $\nu_x$, we treat it as having no regret. Otherwise, we assign it a regret of $R_{\nu_x}(w) - \phi$, the amount by which its regret surpasses $\phi$. This is the same as in formulation (4) except the cost of having regret greater than $\phi$ is simply a constant of 1 in (4). Moreover, this is now a convex problem whenever $g_\nu(x)$ is convex.

**Definition 2 (minimum-violation)** *The* minimum violation *optimization problem is given by*

$$\min_{w \in \mathcal{P}} \mathbb{E}_{\nu_x \sim D_x} \left[ \max\{R_{\nu_x}(w) - \phi, 0\} \right] \tag{5}$$

To gain additional intuition into the choice of this formulation, we can also rewrite this objective as

$$\min_{w \in \mathcal{P}} \mathbb{P}_{\nu_x \sim D_x}(R_{\nu_x}(w) \geq \phi) \cdot \mathbb{E}_{\nu_x \sim D_x} \left[ R_{\nu_x}(w) - \phi \big| R_{\nu_x}(w) \geq \phi \right] \tag{6}$$

Notice that for $\phi = 0$, the above formulation reduces to simply minimizing the expected regret. In addition, for $\phi$ large enough (so that regret is always bounded by $\phi$), the problem becomes fully robust and produces the same solution as (4). In between, this is a combination of two objectives. The left term is the original one in (4) to minimize the probability that regret is larger than $\phi$. The second term is similar to conditional value at risk (CVaR) Rockafellar & Uryasev (2002). The CVaR objective minimizes $\mathbb{E}_{\nu_x}[R_{\nu_x}(w)|R_{\nu_x}(w) \geq q_\alpha(w)]$ where $q_\alpha(w)$ is the $\alpha^{th}$ quantile of the regret distribution of taking decision $w$. To contrast this approach with ours, note that $q_\alpha(w)$ can change for each $w$ while the $\phi$ term remains constant throughout. We see in the computational experiments (section 5.1 and Figure 2) that the CVaR approach produces decisions that change discretely as the robustness parameter changes (the quantile being targeted), whereas the minimum-violation objective produces more continuously changing decisions.

**Overview** The key idea is to coarsen the problem and discretize the feasible region $\mathcal{P}$ into $K$ subsets $H_1, \ldots, H_k$ and determine the probabilities $\mathbb{P}(w^*(\nu_x) \in H_k)$ that the optimal solution $w^*(\nu_x)$ belongs to each $H_k, k = 1, \ldots, K$. We can use these discrete $H_k$ as building blocks to approximate the expectations in (3) and (5). Intuitively, we would like to construct $H_k$ so that, if $w^*(\nu_x)$ belongs to $H_k$, then $R_{\nu_x}(w)$ can be well approximated by some deterministic function of $w$. Then, we minimize the expected regret based off of these individual approximations.

**Discretization** Consider constructing $H_k^\epsilon$ for each datapoint $(x^k, \nu^k)$ corresponding to the set of points whose *regret* is at most $\epsilon$:

$$H_k^\epsilon = \{w \in \mathcal{P} : g_{\nu^k}(w) - g_{\nu^k}(w^*(\nu^k)) \leq \epsilon\} = \{w \in \mathcal{P} : R_{\nu^k}(w) \leq \epsilon\} \tag{7}$$

Now it remains to approximate the probability that $w^*(\nu_x)$ belongs to each $H_k^\epsilon$. We approximate $\mathbb{P}(w^*(\nu_x) \in K_k^\epsilon)$ by leveraging the data we already have for training. For the training data, we compute point estimates of this probability since we have access to the realized cost parameters. That is, for every feature point $x^n$, we can determine whether the optimal decision $w^*(\nu^n)$ either belongs to set $H_k^\epsilon$ or not. For each pair $(x^n, \nu^n)$ and $H_k^\epsilon$, we generate the following labels:

$$p_k^n = \begin{cases} 1, & \text{if } w^*(\nu^n) \in H_k^\epsilon \\ 0, & \text{otherwise} \end{cases} \tag{8}$$

This creates a new multi-label data set $(x^n, (p_k^n)_{k=1,\ldots,N})$. We can then learn a mapping $\hat{p}_k^\epsilon(x)$ which approximates $\mathbb{P}(w^*(\nu_x) \in H_k^\epsilon)$. We accomplish this using any classification method such as for example, logistic regression, decision trees, k-nearest neighbors, and neural networks among others. Figure 1 provides an illustration and the corresponding labels we create.

**Algorithm** We summarize the algorithm as the following steps:

(i) Define subsets $H_k^\epsilon = \{w \in \mathcal{P} : R_{\nu^k}(w) \leq \epsilon\}$ for each datapoint.
(ii) Construct labels $p_k^n$ to indicate whether $w^*(\nu^n) \in H_k^\epsilon$.

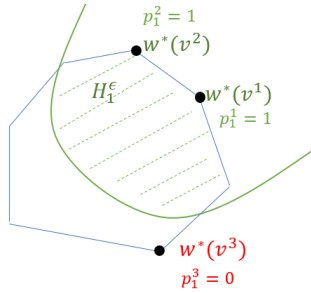

Figure 1: $w^*(\boldsymbol{\nu}^1)$ and $w^*(\boldsymbol{\nu}^2)$ belong to $H_1^\epsilon$, but $w^*(\boldsymbol{\nu}^3)$ does not. Thus, points $x^1, x^2$ are labeled with 1, and $x^3$ is labeled as 0.

(iii) Train ML model $\hat{p}_k^\epsilon(\boldsymbol{x})$ on multi-label dataset $(x^n, (p_k^n)_{k=1,\ldots,N})$.

(iv) For out-of-sample $\boldsymbol{x}$, take decision

$$\hat{w}_{\epsilon,\phi}(\boldsymbol{x}) = \arg\min_{\boldsymbol{w}\in\mathcal{P}} \sum_{k=1}^{N} \hat{p}_k^\epsilon(\boldsymbol{x}) \cdot \max\{R_{\boldsymbol{\nu}^k}(\boldsymbol{w}) - \phi, 0\} \tag{9}$$

This optimization problem in (9) combines the individual predictions of which sets $H_k^\epsilon$ the solution should belong to. To gain some intuition on this last step, consider the case that the sets $H_k^\epsilon$ form a partition of the feasible region $\mathcal{P}$. Note that we will relax this to the general case in Theorem 4.1. Then, conditioning on the events $w^*(\boldsymbol{\nu_x}) \in H_k^\epsilon$, the minimum-violation problem in (5) can be rewritten as

$$\mathbb{E}_{\boldsymbol{\nu_x}\sim D_{\boldsymbol{x}}}\left[\max\{R_{\boldsymbol{\nu_x}}(\boldsymbol{w}) - \phi, 0\}\right] = \sum_k \mathbb{P}(w^*(\boldsymbol{\nu_x}) \in H_\epsilon^k)\mathbb{E}\left[\max\{R_{\boldsymbol{\nu_x}}(\boldsymbol{w}) - \phi, 0\} \mid w^*(\boldsymbol{\nu_x}) \in H_k^\epsilon\right].$$
$$\tag{10}$$

In Theorem 4.1 we will show that the term $\max\{R_{\boldsymbol{\nu}^k}(\boldsymbol{w}) - \phi, 0\}$ approximates the value of $\max\{R_{\boldsymbol{\nu_x}}(\boldsymbol{w}) - \phi, 0\}$, whenever $w^*(\boldsymbol{\nu_x}) \in H_k^\epsilon$. Moreover, these terms are weighted by $\hat{p}_k^\epsilon(\boldsymbol{x})$, the approximations of $\mathbb{P}(w^*(\boldsymbol{\nu_x}) \in H_k^\epsilon)$.

**Alternative interpretation for $\phi = \epsilon$:** We would also like to present an alternative viewpoint of our proposed method which connects the choice of objective $\mathbb{E}[\max\{R_{\boldsymbol{v}}(\boldsymbol{w}) - \phi, 0\}]$ to the rest of the method and how the weights $\hat{p}_k(\boldsymbol{x})$ are generated. This interpretation is unique to our proposed method, and differs from that of Bertsimas & Kallus (2020) and related literature. We can alternatively view the problem as follows. For out-of-sample features $\boldsymbol{x}$, the goal is to find a feasible solution $\boldsymbol{w}$ that best matches the predictions $\hat{p}_k^\epsilon(\boldsymbol{x})$ in terms of which sets $H_k^\epsilon$ the optimal solution $w^*(\boldsymbol{\nu_x})$ should belong to. For example, if we predict that $w^*(\boldsymbol{\nu_x})$ belongs to $H_1^\epsilon$ and to $H_2^\epsilon$ with high probability (meaning weights $\hat{p}_1^\epsilon(\boldsymbol{x}), \hat{p}_2^\epsilon(\boldsymbol{x})$ are high), then the solution we propose should belong to the intersection of sets $H_1^\epsilon$ and $H_2^\epsilon$. The formulation in 9 performs this by implicitly scoring each feasible solution: if a feasible solution $\boldsymbol{w}$ does not belong to $H_k^\epsilon$, we penalize it by our approximation $\hat{p}_k^\epsilon(x)$ that it should have belonged to it multiplied by the distance from $H_k^\epsilon$. However, if $\boldsymbol{w}$ does belong to $H_k^\epsilon$, then there is no penalty. This score exactly corresponds to each term $\hat{p}_k^\epsilon(\boldsymbol{x}) \cdot \max\{R_{\boldsymbol{\nu}^k}(\boldsymbol{w}) - \epsilon, 0\}$. We choose the feasible solution that minimizes overall penalty.

## 4 THEORETICAL REGRET BOUND AND PRACTICAL APPLICATION

**Theorem 4.1** *Under Assumption 3.1, the expected regret of a decision $\boldsymbol{w} \in \mathcal{P}$ can be bounded above by the approximate problem in (9) with probability $1 - \delta$ as follows:*

$$\mathbb{E}\left[\max\{R_{\boldsymbol{\nu_x}}(\boldsymbol{w}) - \theta, 0\}\right] \leq c_\epsilon \cdot \alpha\left(OBJ(\boldsymbol{w}) + M_1\mathcal{E} + \sqrt{\frac{\log 1/\delta}{2N}}\right) \tag{11}$$

*for $\theta = \alpha(\beta_\epsilon + \phi)$ and where $OBJ(\boldsymbol{w}) = \frac{1}{N}\sum_{k=1}^{N} \hat{p}_k(\boldsymbol{x})\max\{R_{\boldsymbol{\nu}^k}(\boldsymbol{w}) - \phi, 0\}$ is the approximation we optimize over in (9). $\mathcal{E}$ is the mean prediction error $\mathcal{E} = \frac{1}{N}\sum_{k=1}^{N} |\hat{p}_k^\epsilon(\boldsymbol{x}) - \mathbb{P}(w^*(\boldsymbol{\nu_x}) \in$*

$H_k^\epsilon)|$ *and constants* $\alpha, \beta_\epsilon$ *depending on the optimization problem. In particular, for the following classes of objectives we have*

1. bi-lipschitz objectives*: for any bi-lipschitz objective, with constants $L, \mu$ such that $\mu \left\| \boldsymbol{w}^1 - \boldsymbol{w}^2 \right\| \leq |g_{\boldsymbol{\nu}}(\boldsymbol{w}^1) - g_{\boldsymbol{\nu}}(\boldsymbol{w}^2)| \leq L \left\| \boldsymbol{w}^1 - \boldsymbol{w}^2 \right\|$, we have $\alpha = L/\mu$ and $\beta_\epsilon = \epsilon$. As an example, this holds for any quadratic optimization problem with bounded feasible region having a positive definite quadratic term.*

2. quantile loss function: *given a prediction $\boldsymbol{w}$ and outcome $\boldsymbol{\nu_x}$, the regression loss for quantile $q$ is given by $g_{\boldsymbol{\nu_x}}(\boldsymbol{w}) = \max\{q(\boldsymbol{\nu_x} - \boldsymbol{w}), (1 - q)(\boldsymbol{w} - \boldsymbol{\nu_x})\}$. Then $\alpha = 1$ and $\beta_\epsilon = \max\{q/(1 - q), (1 - q)/q\}\epsilon$. For example, this is also applicable to the inventory stock problem in section 5.1.*

*Furthermore, $c_\epsilon$ is a constant factor describing for any $x$, how often $w^*(\boldsymbol{\nu_x})$ will have regret at most $\epsilon$ with respect to a random other cost vector. In particular, $c_\epsilon = 1/\min_{\boldsymbol{x}} \mathbb{P}_{\boldsymbol{y}}(R_{\boldsymbol{\nu_y}}(w^*(\boldsymbol{\nu_x})) \leq \epsilon)$.*

The full proof can be found in Appendix A. Moreover, we prove stability of the output of our proposed model under perturbations in the data. This can be found in the appendix (see Appendix B). In short, the stability result describes the change in the decision $\hat{w}_{\epsilon,0}(\boldsymbol{x})$ when the dataset is perturbed by noise. If each of the learning algorithms used to train $\hat{p}_k(\boldsymbol{x})$ have hypothesis stability (defined in the appendix, see definition 3), then the output $\hat{w}_{\epsilon,0}(\boldsymbol{x})$ also changes by a small amount when the dataset is perturbed. Before the computational section, we discuss some of the main practical issues and take-aways from Theorem 4.1. In particular, we discuss how to practically choose $\epsilon$ and how this affects each term in the bound in theorem 4.1.

**Discussion on choosing $\epsilon$.** By definition, $\epsilon$ determines the size of sets $H_k^\epsilon$. This in turn affects the resulting multi-labelling $p_k^n, k = 1, \ldots, N$. If for example, the sets $H_k^\epsilon$ do not intersect, then each vector $\boldsymbol{p}^n = (p_k^n)_{k=1,\ldots,N}$ has a single non-zero entry. This would make $\hat{p}_k(\boldsymbol{x})$ impossible to learn.

As such, we propose the following method of choosing $\epsilon$: *choose $\epsilon$ large enough so that for each vector $\boldsymbol{p}^n$, at least some fraction, which we denote by $\gamma_\epsilon$, of its entries are non-zero.* Then, from Theorem 4.1, we argue that **given $N$ datapoints, one should choose $\epsilon$ large enough so that $\gamma_\epsilon \geq 1/\sqrt{N}$**. We can see this as follows by viewing the impact of $\epsilon$ on each term of the bound:

*Effect on $c_\epsilon$.* Recall that if $p_k^n = 1$, this is equivalent to $R_{\boldsymbol{\nu^k}}(w^*(\boldsymbol{\nu^n})) \leq \epsilon$. Therefore, a $\gamma_\epsilon$ chosen in this way implies that

$$\mathbb{P}_{\boldsymbol{y}}(R_{\boldsymbol{v_y}}(w^*(\boldsymbol{v_x})) \leq \epsilon) \approx \gamma_\epsilon. \tag{12}$$

Moreover, this results in $c_\epsilon \approx 1/\gamma_\epsilon$.

*Effect on $\mathcal{E}$.* $\gamma_\epsilon$ also affects the error of the ML models, namely $\mathcal{E}$. For instance, labelling everything with a zero will have an error of $\mathcal{E} = \gamma_\epsilon$. In general, any ML model that improves beyond this baseline will have $\mathcal{E} \leq \gamma_\epsilon$.

*Effect on $OBJ(\boldsymbol{w})$.* We can bound $OBJ(\boldsymbol{w})$ by

$$OBJ(\boldsymbol{w}) = \frac{1}{N} \sum_{k=1}^{N} \hat{p}_k^\epsilon(\boldsymbol{x}) \max\{R_{\boldsymbol{\nu^k}}(\boldsymbol{w}) - \phi, 0\} \tag{13}$$

$$\leq M_1 \frac{1}{N} \sum_{k=1}^{N} \hat{p}_k(\boldsymbol{x}) \leq M_1 \left( \mathcal{E} + \frac{1}{N} \sum_{k=1}^{N} \mathbb{P}(w^*(\boldsymbol{\nu_x}) \in H_k^\epsilon). \right) \tag{14}$$

Finally, the term $\frac{1}{N} \sum_{k=1}^{N} \mathbb{P}(w^*(\boldsymbol{\nu_x}) \in H_k^\epsilon)$ concentrates around its expectation which is $\gamma_\epsilon \approx \mathbb{P}_{\boldsymbol{y}}(R_{\boldsymbol{\nu_y}}(w^*(\boldsymbol{\nu_x})) \leq \epsilon)$. It follows that $OBJ(\boldsymbol{w}) \lessapprox 2M_1\gamma_\epsilon$, given that $\mathcal{E} \leq \gamma_\epsilon$ as argued previously.

Overall, this implies that the right hand side of the bound in theorem 4.1 has $c_\epsilon$ on the order of $1/\gamma_\epsilon$, while $OBJ(\boldsymbol{w})$ and $\mathcal{E}$ are both on the order of $\gamma_\epsilon$. Putting these together, we see that these cancel out to be overall on the order of $O(M_1)$. The only remaining term is $c_\epsilon \cdot \sqrt{\log(1/\delta)/2N}$. Hence, we need enough data so that this term also becomes on the order of $O(1)$. Therefore, we need on the order of $1/\gamma_\epsilon^2$ datapoints or that $\gamma_\epsilon \geq 1/\sqrt{N}$. We present results in the computational experiments section about the effect of $\epsilon$ on the quality of decisions produced.

The remaining question is now, how small can $\epsilon$ be so that $\gamma_\epsilon \geq 1/\sqrt{N}$? In practice, this can be determined on a case-by-case basis for each dataset, for instance by using binary search. Of course, $\gamma_\epsilon \geq 1/\sqrt{N}$ is only a rough guide to understand the magnitude needed. A process like cross-validation can ultimately be used to choose $\epsilon$. Theoretically, in the worst case either (1) such an $\epsilon$ needs to be a constant fraction of the diameter of the feasible region or (2) $\epsilon$ is kept small but we need an amount of data that is exponential in the dimension of the decisions. However, this is often not the case in practice when presented with real-world data. Ultimately this largely depends on the distribution of the decisions $w^*(\boldsymbol{\nu})$ and $\mathcal{P}$ itself. In real-world data, the distribution of $w^*(\boldsymbol{\nu})$ would not pathologically uniformly cover the feasible region, but rather be more clustered around smaller regions. Moreover, the feasible region itself plays a crucial role. Consider the case that decisions $\boldsymbol{w} \in \mathbb{R}^d$ are constrained to a feasible region $\mathcal{P}$ that is an $s$-dimensional subspace of the ambient space $\mathbb{R}^d$. For instance, if $\mathcal{P} = \{w : Aw = b, w \geq 0\}$ where the null space of $A$ has dimension $s$.

## 5 COMPUTATIONAL RESULTS

We consider four applications of the proposed approach. Two applications come from Donti et al. (2017): a synthetic inventory stock problem and a real-world energy scheduling task. We show that this discretization approach is competitive with other methods in terms of expected cost, and has significant improvement in terms of robustness with up to 20 times lower cost in worst-case scenarios. Moreover, we also compare against other non-contextual robust optimization methods. Due to space limitations, we present this last experiment in appendix C.1. We also put into practice the earlier discussion regarding the choice of $\epsilon$ and how its value affects the quality of the decisions ranging from performing better at minimizing the expected cost or being robust.

### 5.1 INVENTORY STOCK PROBLEM

Consider the classical newsvendor problem in which a given product has uncertain demand $d$ as well as observed covariates $\boldsymbol{x}$. Each day we observe the new features $\boldsymbol{x}$ and must make a decision $w$ for the amount of product to supply. Afterwards, the true demand is realized. For each unit of supply above the demand, there is a unit cost of $c_h$ (for holding an item overnight in the store) and for each unit of supply below the demand, we incur a backorder or lost sales unit cost $c_b$ (for example, cost of expedited shipping to ensure the product arrives the next day). It has been shown that the optimal order quantity is the $c_b/(c_b + c_h)$ quantile of the demand distribution (see Arrow et al. (1951)). In principle, one could apply the quantile loss function to predict this quantity and solve the problem in a two-stage manner. However, to consider the identical problem also presented in Donti et al. (2017), we consider a version with additional quadratic costs to over or under-stocking as well as an ordering cost. The objective is given as

$$g_d(w) = c_0 w + c_q w^2 + c_b \max\{d - w, 0\} + \frac{1}{2} q_b \max\{d - w, 0\}^2 + \tag{15}$$

$$c_h \max\{w - d, 0\} + \frac{1}{2} q_h \max\{w - d, 0\}^2 \tag{16}$$

**Experimental setup:** We use the same unit cost parameters as well as data, and compare against the same models as in Donti et al. (2017). However, we present not only the average cost incurred on the testing data but also on various quantiles of the cost distribution. We plot on the $x$-axis the mean cost incurred by each method, and on the $y$-axis the cost at the $q^{th}$ quantile.

The problem of $\min_{w \in \mathcal{P}} \mathbb{E}\left[\max\{R_{\boldsymbol{\nu}_x}(\boldsymbol{w}) - \phi, 0\}\right]$ can also be solved by existing methods such as Bertsimas & Kallus (2020) by thinking of the objective function not as $g_{\boldsymbol{\nu}}(\boldsymbol{x})$ but rather as $\max\{R_{\boldsymbol{\nu}_x}(\boldsymbol{w}) - \phi, 0\}$ directly. In this approach, weights are generated by K-nearest neighbors (or other ML models like linear or tree models). However, these methods do not generate these weights based on the optimization problem itself. In contrast our proposed method generates weights explicitly depending on the optimization task. We denote this approach as *KNN + minimum-violation* in the experiments. We also compare against the method of Bertsimas & Kallus (2020) where we use CVaR as the objective, namely $\mathbb{E}[R_{\boldsymbol{\nu}_x}(\boldsymbol{w})|R_{\boldsymbol{\nu}_x}(\boldsymbol{w}) \geq q_\alpha(R_{\boldsymbol{\nu}_x}(\boldsymbol{w}))]$, where $q_\alpha(Z)$ is the $\alpha$-quantile of a random variable $Z$. We consider a range of values for the quantile $\alpha$ in the experiments. In addition, we also compare against another data-driven contextual robust optimization methods where one solves the following problem. For an out-of-sample $\boldsymbol{x}$, find the $K$ nearest neighbors in the data,

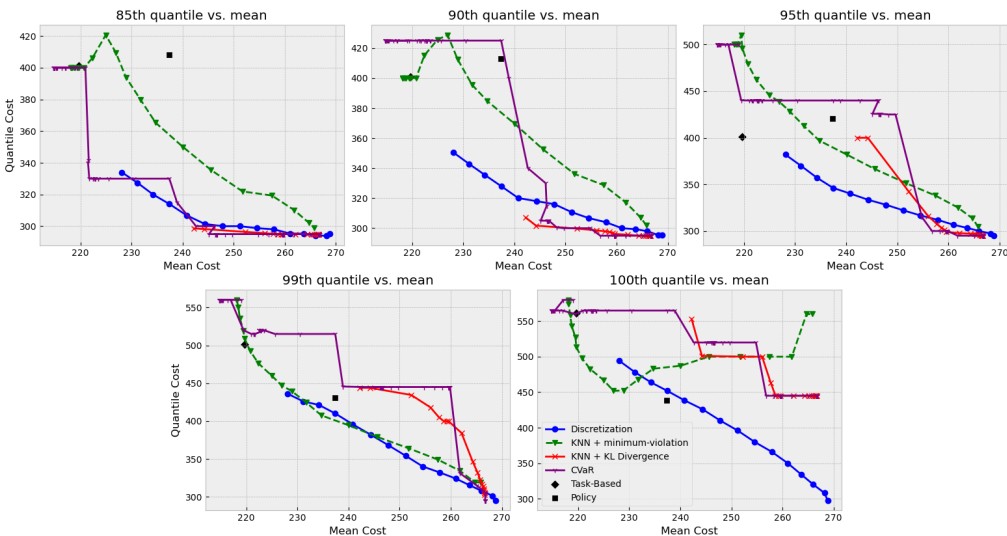

Figure 2: Comparison of average cost of each method vs. $85^{th}, 90^{th}, 95^{th}, 99^{th}$ and $100^{th}$ quantile cost. We vary the degree of robustness for each method. As this robustness parameter increases, the mean cost increases, but the quantile cost (generally) decreases for all methods.

namely, $\mathcal{N}(\boldsymbol{x}, K)$. Unlike Bertsimas & Kallus (2020) which minimizes the average cost, we weight the remaining $K$ datapoints adversarially according to a distribution $\pi$ that is, within a Kullback-Leibler divergence distance of at most $r$ from a uniform distribution, assigning weights $1/K$ to each datapoint (so that $D_{KL}(\pi||1/K) \leq r$). Due to space limitations, details and formulations for each of these approaches can be found in appendix C.1. Moreover, we compare against the method in Donti et al. (2017). This is an end-to-end method which trains a neural network to predict a discrete probability distribution for the possible values of demand. We refer to this as the *task-based* method in Figure 2. This work makes use of the OptNet framework in Amos & Kolter (2017) to compute gradients of the loss function with respect to the predicted demand $d$. Furthermore, we also compare against a *policy* optimizer approach. Here, one does not make a forecast for demand, but rather the neural network model directly outputs the policy/decision to take.

Each of these methods use a linear model to make predictions (or decisions, as in the case of the policy optimizer). For the other weight-based methods (KNN + KL divergence and those based on Bertsimas & Kallus (2020)) we use a KNN method to generate weights (with $K = 10$). For consistency, our approach also uses a KNN model to predict the $\hat{p}_k(\boldsymbol{x})$, also using $K = 10$ neighbors. Each $\hat{p}_k(\boldsymbol{x})$ predicts a value between 0 and 1 based on the average label assigned to the $K$-nearest neighbors of $\boldsymbol{x}$ in the training data. This is done independently for each $k$. In contrast, for the following experiment in section 5.2, the models $\hat{p}_k(\cdot)$ are trained simultaneously by a neural network.

**Results:** In Figure 2 we report the mean and the $85^{th}, 90^{th}, 95^{th}, 99^{th}$ and $100^{th}$ quantile (out-of-sample) cost of each decision made by the approaches as we vary the level of robustness for each approach ($\phi$ for the minimum-violation objective and $\alpha$ in for CVaR). The *KNN + minimum-violation* method and our proposed method have the same objective, but use different methods of solving it. We see that both approaches produce similarly shaped mean vs. quantile cost curves but the discretization method consistently has lower $q^{th}$ quantile cost for the same average cost for all $q = 85$ to $q = 100$. As this quantile approaches 100, the gap between the two decreases but of note, at $q = 100$, all approaches, other than the discretization method performs poorly, worse than even the policy method in terms of robustness. The CVaR and traditional robust methods also deteriorate in performance as the quantile increases.

## 5.2 LOAD FORECASTING AND GENERATOR SCHEDULING

Next, we consider a real-world problem for generator scheduling using 8 years of real electrical grid data from PJM, an electricity routing company coordinating the movement of electricity throughout

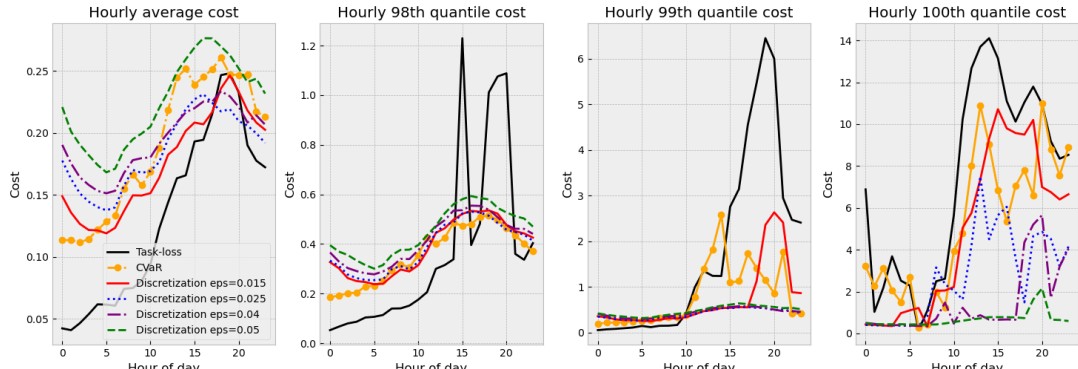

Figure 3: Reporting for each hour of the day the mean, $99^{th}$ quantile, and maximum cost of each method. A fixed $\phi = 0.5$ value for the discretization method was used for all results.

13 states. We use the same range of data as also used in Donti et al. (2017). Here, we must make decisions $\boldsymbol{w} \in \mathbb{R}^{24}$ for the amount of electricity generation for each hour of the following day. Similar to the inventory problem, the operator incurs a cost $\gamma_e$ for excess generation and a cost $\gamma_s$ for a shortage in generation. In addition, power plants have physical limitations prohibiting large changes in generation from one hour to the next. The objective and constraints are given as:

$$g_{\boldsymbol{d}}(\boldsymbol{w}) = \sum_{i=1}^{24} \gamma_s \max\{\boldsymbol{d}_i - \boldsymbol{w}_i, 0\} + \gamma_e \max\{\boldsymbol{w}_i - \boldsymbol{d}_i, 0\}, \qquad |w_{i+1} - w_i| \le r, \ i = 1, \dots 23 \quad (17)$$

**Experimental setup:** We use the same setup used in Donti et al. (2017), using the same parameters for the problem as well as data. We make use of the same data preprocessing and feature engineering. They use a two-layer (each layer of width 200) network with an additional residual connection from the inputs to the outputs. We use the same architecture to learn the labeling $\hat{p}_k^\epsilon(\boldsymbol{x})$. In addition, we compare against a cost-weighted model minimizing mean-squared error which periodically reweights training samples based on their task-based cost. Finally, we also compare against the method described in the previous section with objective to minimize CVaR.

**Results:** We compare the average cost as well as the $98 - 100^{th}$ quantiles of the cost distribution for each method. In particular, we also present results for different choices of $\epsilon$. Following the discussion of theorem 4.1, a starting point would be to choose $\epsilon$ so that the average number of positive labels , $\gamma_\epsilon$, is around $1/\sqrt{N}$. We choose different epsilon so that $\gamma_\epsilon = 0.015, 0.025, 0.04, 0.05$ where $1/\sqrt{N} \approx 0.02$ (here we have $N = 2,553$ training points). As $\epsilon$ and $\gamma_\epsilon$ decrease, we find that the solutions better target minimizing the expected cost, while increasing $\epsilon$ will improve performance on higher quantiles of the cost distribution. In particular, when $\gamma_\epsilon = 0.025$, the method outperforms even the task-based method on average cost at peak demand hours of the day (hours 15-22). At the other extreme at $\epsilon = 0.05$, we find that the worst case cost across the entire day is nearly constant and up to more than 20 times lower than the CVaR and other methods. However, setting $\epsilon = 0.01$ to be too small does not introduce enough robustness. While it performs best in terms of average cost, its worst-case cost spikes suddenly.

## 6 CONCLUSIONS

We proposed a novel method for contextual stochastic optimization based off of discretizing the feasible region into subsets and learning how the optimal solution maps to each subset. We proved analytical guarantees on the bound between the expected out-of-sample cost compared to the approximate objective proposed in 9. Finally, we present computational experiments on three datasets, including a real-world electricity generation problem, and show our proposed method is competitive against other end-to-end approaches and provides significantly more robust solutions, even when compared to other robust optimization methods. Future directions of research may include devising different constructions of the subsets $H_k^\epsilon$ and to consider uncertainty in constraints as well.

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

# A PROOF OF THEOREM 1

**Theorem 1:** Under basic boundedness assumptions on the optimization problem (see Assumption 3.1), the expected regret of a decision $\boldsymbol{w} \in \mathcal{P}$ can be bounded above by the approximate problem in 9 with probability $1 - \gamma$ as follows:

$$\mathbb{E}\left[(R_{\boldsymbol{\nu_x}}(\boldsymbol{w}) - \theta) \cdot \mathbb{1}(R_{\boldsymbol{\nu_x}}(\boldsymbol{w}) \geq \theta)\right] \leq c_\epsilon \cdot \alpha \left(OBJ(\boldsymbol{w}) + M_1 \mathcal{E} + \sqrt{\frac{\log 1/\gamma}{2N}}\right) \tag{18}$$

for $\theta = \alpha\beta_\epsilon + \alpha\phi$ and where $OBJ(\boldsymbol{w}) = \frac{1}{N}\sum_{k=1}^{N} \hat{p}_k^\epsilon(\boldsymbol{x}) \max\{R_{\boldsymbol{\nu^k}}(\boldsymbol{w}) - \phi, 0\}$ is the approximation we optimize over in 9. $\mathcal{E}$ is the mean prediction error $\mathcal{E} = \frac{1}{N}\sum_{k=1}^{N} |\hat{p}_k^\epsilon(\boldsymbol{x}) - \mathbb{P}(w^*(\boldsymbol{\nu_x}) \in H_k)|$ and constants $\alpha, \beta_\epsilon$ depending on the optimization problem. In particular, for the following classes of objectives we have

1. *bi-lipschitz objectives*: for any bi-lipschitz objective, with constants $L, \mu$ such that $\mu \left\|\boldsymbol{w}^1 - \boldsymbol{w}^2\right\| \leq |g_{\boldsymbol{\nu}}(\boldsymbol{w}^1) - g_{\boldsymbol{\nu}}(\boldsymbol{w}^2)| \leq L \left\|\boldsymbol{w}^1 - \boldsymbol{w}^2\right\|$, we have $\alpha = L/\mu$ and $\beta_\epsilon = \epsilon$. As an example, this holds for any quadratic optimization problem with bounded feasible region having a positive definite quadratic term.

2. *quantile loss function:* given a prediction $\boldsymbol{w}$ and outcome $\boldsymbol{\nu_x}$, the regression loss for quantile $q$ is given by $g_{\boldsymbol{\nu_x}}(\boldsymbol{w}) = \max\{q(\boldsymbol{\nu_x} - \boldsymbol{w}), (1-q)(\boldsymbol{w} - \boldsymbol{\nu_x})\}$. Then $\alpha = 1$ and $\beta_\epsilon = \max\{q/(1-q), (1-q)/q\}\epsilon$.

Furthermore, $c_\epsilon$ is a constant factor describing for any $x$, how often $w^*(\boldsymbol{\nu_x})$ will have regret at most $\epsilon$ with respect to a random other cost vector. In particular, $c_\epsilon = 1/\min_{\boldsymbol{x}} \mathbb{P}_{\boldsymbol{y}}(R_{\boldsymbol{\nu_y}}(w^*(\boldsymbol{\nu_x})) \leq \epsilon)$.

**Proof:** We prove this statement for a broader class of objectives. In particular, for functions having the following condition: whenever $R_{\boldsymbol{\nu_y}}(w^*(\boldsymbol{\nu_x})) \leq \epsilon$ (where $\boldsymbol{y}$ is a feature vector like $\boldsymbol{x}$), then we must have that $R_{\boldsymbol{\nu_x}}(\boldsymbol{w}) \leq \alpha R_{\boldsymbol{\nu_y}}(\boldsymbol{w}) + \beta_\epsilon$. We will then show this holds for instance for bi-lipschitz and quantile loss functions. We proceed to prove this as follows. We first consider our formulation in 9 when the approximations $\hat{p}_k^\epsilon(\boldsymbol{x})$ are exact. Let

$$OBJ^*(\boldsymbol{w}) = \frac{1}{N}\sum_{k=1}^{N} \mathbb{P}(w^*(\boldsymbol{\nu_x} \in H_k^\epsilon) \max\{g_{\boldsymbol{\nu^k}}(\boldsymbol{w}) - \phi, 0\} \tag{19}$$

We will show that

$$c \cdot \alpha \left(OBJ^*(\boldsymbol{w}) + \sqrt{\frac{\log 1/\delta}{2N}}\right) \geq \mathbb{E}_{\boldsymbol{x}}\left[\max\{R_{\boldsymbol{\nu_x}}(\boldsymbol{w}) - \alpha(\phi + \beta_\epsilon), 0\}\right] \tag{20}$$

Then, we prove the general statement for $OBJ(\boldsymbol{w})$ by showing that $|OBJ(\boldsymbol{w}) - OBJ^*(\boldsymbol{w})| \leq M_1/N \sum_{k=1}^{N} |\hat{p}_k^\epsilon(\boldsymbol{x}) - \mathbb{P}(w^*(\boldsymbol{\nu_x}) \in H_k)|$.

First, we show 20. We have

$$OBJ^*(\boldsymbol{w}) = \frac{1}{N}\sum_{k=1}^{N} \mathbb{P}(w^*(\boldsymbol{\nu_x}) \in H_k^\epsilon) \max\{g_{\boldsymbol{\nu^k}}(\boldsymbol{w}) - \phi, 0\} \tag{21}$$

$$= \mathbb{E}_{\boldsymbol{x}}\left[\frac{1}{N}\sum_{k=1}^{N} \mathbb{1}(R_{\boldsymbol{\nu^k}}(w^*(\boldsymbol{\nu_x}) \leq \epsilon)) \max\{g_{\boldsymbol{\nu^k}}(\boldsymbol{w}) - \phi, 0\}\right] \tag{22}$$

Next, we can treat the $\boldsymbol{\nu}^k$ as realizations of a random variable as well. The inner term in the expectation, $\mathbb{E}_{\boldsymbol{x}}[\frac{1}{N}\sum_{k=1}^{N} \mathbb{1}(R_{\boldsymbol{\nu^k}}(w^*(\boldsymbol{\nu_x}) \leq \epsilon)) \max\{g_{\boldsymbol{\nu^k}}(\boldsymbol{w}) - \phi, 0\}]$, is concentrated around its mean, $\mu$. The $\boldsymbol{\nu}^k$ are identically distributed and independent. Therefore, so is a function of these random variables. Let $\boldsymbol{y}$ be a random variable from the same distribution as all the $\boldsymbol{\nu}^k$. Then, by Hoeffding's inequality,

$$\mathbb{P}_{\boldsymbol{y}}\left(\frac{1}{N}\sum_{k=1}^{N} \mathbb{P}(w^*(\boldsymbol{\nu_x}) \in H_k^\epsilon) \max\{g_{\boldsymbol{\nu^k}}(\boldsymbol{w}) - \phi, 0\} - \mu \geq t\right) \leq \exp\left(-2Nt^2\right) \tag{23}$$

Therefore, with probability $1 - \delta$,

$$OBJ^*(\boldsymbol{w}) \geq \mathbb{E}_{\boldsymbol{x}} \left[ \mathbb{E}_{\boldsymbol{y}} \left[ \mathbb{P}(R_{\boldsymbol{\nu}_{\boldsymbol{y}}}(w^*(\boldsymbol{\nu}_{\boldsymbol{x}})) \leq \epsilon) \max\{g_{\boldsymbol{\nu}_{\boldsymbol{y}}}(\boldsymbol{w}) - \phi, 0\} \right] \right] - \sqrt{\frac{\log 1/\delta}{2N}} \quad (24)$$

with $\boldsymbol{y}$ a random variable identically distributed like the $\boldsymbol{\nu}^k$. Rewriting the expectation over $\boldsymbol{y}$ by conditioning over the event $R_{\boldsymbol{\nu}_{\boldsymbol{y}}}(w^*(\boldsymbol{\nu}_{\boldsymbol{x}})) \leq \epsilon$, we have

$$OBJ^*(\boldsymbol{w}) \geq \mathbb{E}_{\boldsymbol{x}} \left[ \mathbb{P}_{\boldsymbol{y}}(R_{\boldsymbol{\nu}_{\boldsymbol{y}}}(w^*(\boldsymbol{\nu}_{\boldsymbol{x}})) \leq \epsilon) \mathbb{E}_{\boldsymbol{y}}[\max\{g_{\boldsymbol{\nu}_{\boldsymbol{y}}}(\boldsymbol{w}) - \phi, 0\} \mid R_{\boldsymbol{\nu}_{\boldsymbol{y}}}(w^*(\boldsymbol{\nu}_{\boldsymbol{x}})) \leq \epsilon] \right] - \sqrt{\frac{\log 1/\delta}{2N}}$$
$$(25)$$

By assumption, we simply bound the term $\mathbb{P}_{\boldsymbol{y}}(R_{\boldsymbol{\nu}_{\boldsymbol{y}}}(w^*(\boldsymbol{\nu}_{\boldsymbol{x}})) \leq \epsilon) \geq 1/c$. And again by assumption on the structure of the objective function, we can bound

$$\mathbb{E}_{\boldsymbol{y}}[\max\{g_{\boldsymbol{\nu}_{\boldsymbol{y}}}(\boldsymbol{w}) - \phi, 0\} \mid R_{\boldsymbol{\nu}_{\boldsymbol{y}}}(w^*(\boldsymbol{\nu}_{\boldsymbol{x}})) \leq \epsilon] \geq \max\{\frac{1}{\alpha} R_{\boldsymbol{\nu}_{\boldsymbol{x}}}(\boldsymbol{w}) - \beta_\epsilon - \phi, 0\} \quad (26)$$

Combining this with 25, and rearranging, shows that with probability $1 - \delta$,

$$c \cdot \alpha \left( OBJ^*(\boldsymbol{w}) + \sqrt{\frac{\log 1/\delta}{2N}} \right) \geq \mathbb{E}_{\boldsymbol{x}} \left[ \max\{R_{\boldsymbol{\nu}_{\boldsymbol{x}}}(\boldsymbol{w}) - \alpha\beta_\epsilon - \alpha\phi, 0\} \right] \quad (27)$$

Next, we show that $|OBJ(\boldsymbol{w}) - OBJ^*(\boldsymbol{w})| \leq \mathcal{E} = M_1/N \sum_{k=1}^{N} |\hat{p}_k^\epsilon(\boldsymbol{x}) - \mathbb{P}(w^*(\boldsymbol{\nu}_{\boldsymbol{x}}) \in H_k)|$. We have

$$|OBJ(\boldsymbol{w}) - OBJ^*(\boldsymbol{w})| \leq \frac{1}{N} \sum_{k=1}^{N} (\hat{p}_k^\epsilon(\boldsymbol{x}) - \mathbb{P}(w^*(\boldsymbol{\nu}_{\boldsymbol{x}}) \in H_k)) R_{\boldsymbol{\nu}^k}(\boldsymbol{w}) \quad (28)$$

By assumption 3.1 $R_{\boldsymbol{\nu}^k}(\boldsymbol{w}) \leq M_1$ and hence the claim follows. Finally, it remains to show the initial conditions hold.

1. *bi-lipschitz objectives:* For any $\boldsymbol{x}, \boldsymbol{y}$, we have

$$g_{\boldsymbol{\nu}_{\boldsymbol{x}}}(\boldsymbol{w}) - g_{\boldsymbol{\nu}_{\boldsymbol{x}}}(w^*(\boldsymbol{\nu}_{\boldsymbol{x}})) \leq L \, \|\boldsymbol{w} - w^*(\boldsymbol{\nu}_{\boldsymbol{x}})\| \quad (29)$$

$$\leq \frac{L}{\alpha} |g_{\boldsymbol{\nu}_{\boldsymbol{y}}}(\boldsymbol{w}) - g_{\boldsymbol{\nu}_{\boldsymbol{y}}}(w^*(\boldsymbol{\nu}_{\boldsymbol{x}}))| \quad (30)$$

$$\leq \frac{L}{\alpha} |g_{\boldsymbol{\nu}_{\boldsymbol{y}}}(\boldsymbol{w}) - g_{\boldsymbol{\nu}_{\boldsymbol{y}}}(w^*(\boldsymbol{\nu}_{\boldsymbol{y}})) + g_{\boldsymbol{\nu}_{\boldsymbol{y}}}(w^*(\boldsymbol{\nu}_{\boldsymbol{y}})) - g_{\boldsymbol{\nu}_{\boldsymbol{y}}}(w^*(\boldsymbol{\nu}_{\boldsymbol{x}}))| \quad (31)$$

$$\leq \frac{L}{\alpha} |g_{\boldsymbol{\nu}_{\boldsymbol{y}}}(\boldsymbol{w}) - g_{\boldsymbol{\nu}_{\boldsymbol{y}}}(w^*(\boldsymbol{\nu}_{\boldsymbol{y}}))| + \epsilon \quad (32)$$

The last inequality follows by noting that by definition $g_{\boldsymbol{\nu}_{\boldsymbol{y}}}(\boldsymbol{w}) - g_{\boldsymbol{\nu}_{\boldsymbol{y}}}(w^*(\boldsymbol{\nu}_{\boldsymbol{y}})) = R_{\boldsymbol{\nu}_{\boldsymbol{y}}}(w^*(\boldsymbol{\nu}_{\boldsymbol{x}})) \leq \epsilon$. So indeed,

$$R_{\boldsymbol{\nu}_{\boldsymbol{x}}}(\boldsymbol{w}) \leq \frac{L}{\alpha} R_{\boldsymbol{\nu}_{\boldsymbol{y}}}(\boldsymbol{w}) + \epsilon \quad (33)$$

2. *quantile loss function:* First, note that $R_{\boldsymbol{\nu}_{\boldsymbol{x}}}(\boldsymbol{w}) = g_{\boldsymbol{x}}(\boldsymbol{w})$ and $w^*(\boldsymbol{\nu}_{\boldsymbol{x}}) = \boldsymbol{\nu}_{\boldsymbol{x}}$ for any $\boldsymbol{x}, \boldsymbol{w}$. Further, note that the condition $R_{\boldsymbol{\nu}_{\boldsymbol{y}}}(w^*(\boldsymbol{\nu}_{\boldsymbol{x}})) \leq \epsilon$ then implies that

$$\max\{q(\boldsymbol{\nu}_{\boldsymbol{y}} - \boldsymbol{\nu}_{\boldsymbol{x}}), (1 - q)(\boldsymbol{\nu}_{\boldsymbol{x}} - \boldsymbol{\nu}_{\boldsymbol{y}})\} \leq \epsilon \quad (34)$$

$$|\boldsymbol{\nu}_{\boldsymbol{x}} - \boldsymbol{\nu}_{\boldsymbol{y}}| \leq \min \left\{ \frac{1}{q}, \frac{1}{1 - q} \right\} \epsilon \quad (35)$$

Moreover, $R_{\boldsymbol{\nu}_{\boldsymbol{x}}}(\boldsymbol{w}) = g_{\boldsymbol{\nu}_{\boldsymbol{x}}}(\boldsymbol{w}) = g_{\boldsymbol{\nu}_{\boldsymbol{x}} - \boldsymbol{\nu}_{\boldsymbol{y}} + \boldsymbol{\nu}_{\boldsymbol{y}}}(\boldsymbol{w})$. So,

$$R_{\boldsymbol{\nu}_{\boldsymbol{x}}}(\boldsymbol{w}) = \max\{q(\boldsymbol{\nu}_{\boldsymbol{x}} - \boldsymbol{\nu}_{\boldsymbol{y}} + \boldsymbol{\nu}_{\boldsymbol{y}} - \boldsymbol{w}) + (1 - q)(\boldsymbol{w} - (\boldsymbol{\nu}_{\boldsymbol{x}} - \boldsymbol{\nu}_{\boldsymbol{y}} + \boldsymbol{\nu}_{\boldsymbol{y}}))\} \quad (36)$$

$$\leq g_{\boldsymbol{\nu}_{\boldsymbol{y}}}(\boldsymbol{w}) + \max\{q, 1 - q\}|\boldsymbol{\nu}_{\boldsymbol{x}} - \boldsymbol{\nu}_{\boldsymbol{y}}| \quad (37)$$

$$\leq g_{\boldsymbol{\nu}_{\boldsymbol{y}}}(\boldsymbol{w}) + \max\{q, 1 - q\} \min \left\{ \frac{1}{q}, \frac{1}{1 - q} \right\} \epsilon \quad (38)$$

$$\leq g_{\boldsymbol{\nu}_{\boldsymbol{y}}}(\boldsymbol{w}) + \max \left\{ \frac{1 - q}{q}, \frac{q}{1 - q} \right\} \epsilon \quad (39)$$

## B    STABILITY OF DECISIONS UNDER DATA UNCERTAINTY

In this section we consider the change in output $\hat{w}_{\epsilon,0}(\boldsymbol{x})$ when the training dataset changes under noise. In particular, we define stability as follows. Let $Z = \{\boldsymbol{\nu}^n\}_{n=1}^N$ and let $\tilde{Z} = \{\boldsymbol{\nu}^n + \delta^n\}_{n=1}^N$ be a perturbation of the dataset $Z$ by noise $\delta^n$ with bounded norm $\|\delta^n\|_2 \leq \alpha$. We wish to bound the difference in decisions when training under $Z$ and any noisy perturbation $\tilde{Z}$. We first make the following assumption regarding the algorithm used learn the labelling $\hat{p}$:

**Definition 3 (Hypothesis Stability)** *Let $p^Z(\boldsymbol{x}) \in [0,1]$ be the prediction output of a learning algorithm trained on a dataset $Z$ of size $N$ for a binary classification problem. We say $p^Z$ has hypothesis stability $\beta_N$ if*

$$\forall i = 1, \ldots, N, \quad \mathbb{E}_x \left| p^Z(\boldsymbol{x}) - p^{Z^{\setminus i}}(\boldsymbol{x}) \right| \leq \beta_N, \tag{40}$$

*where $Z^{\setminus i}$ is the same data set as $Z$, but with the $i^{th}$ point removed. If $\beta_N$ decreases at a rate of $O(1/N)$ with data size $N$, the algorithm is called* stable.

Note that this is a significantly different notion of stability — it only accounts for the addition of a single new datapoint, whereas in our setting every datapoint is perturbed by noise. Moreover, many learning algorithms have this stability property. For example, support vector machines and regularized least squares (Bousquet & Elisseeff (2000)), K-NN classifiers with a $\{0,1\}$ loss function (Devroye & Wagner (1979)), and all learning algorithms (such as neural networks) with Tikhonov regularization (Roscasco & Poggio (2009)).

**Assumption B.1** *The learning algorithm to train and generate each predictor $\hat{p}_k(\cdot)$ has hypothesis stability with rate $\beta_{N,k} = O(1/N)$ where $N$ is the size of the dataset.*

We also make the following assumptions about the optimization problem itself:

**Assumption B.2** *The function $g_{\boldsymbol{\nu}}(\boldsymbol{w})$ is linear with respect to the uncertain parameters $\boldsymbol{\nu}$, although it may be nonlinear with respect to $\boldsymbol{w}$. Moreover, $g_{\boldsymbol{\nu}}(\boldsymbol{w})$ is $L_1$-Lipschitz with respect to $\boldsymbol{w}$ for any $\boldsymbol{\nu}$. So, for $\boldsymbol{w}_1, \boldsymbol{w}_2$:*

$$|g_{\boldsymbol{\nu}}(\boldsymbol{w}_1) - g_{\boldsymbol{\nu}}(\boldsymbol{w}_2)| \leq L_1 \|\boldsymbol{w}_1 - \boldsymbol{w}_2\| \tag{41}$$

*and $w^*(\boldsymbol{\nu})$ is $L_2$-Lipschitz with respect to $\boldsymbol{\nu}$. That is, for $\boldsymbol{\nu}^1, \boldsymbol{\nu}^2$,*

$$\left\| w^*(\boldsymbol{\nu}^1) - w^*(\boldsymbol{\nu}^2) \right\| \leq L_2 \left\| \boldsymbol{\nu}^1 - \boldsymbol{\nu}^2 \right\| \tag{42}$$

For example, any convex unconstrained minimization problem with Lipschitz objective function. Additionally, quadratic optimization problems with any linear constraints also satisfy these assumptions (see for instance Coroianu (2016)).

**Theorem B.1** *Let $\hat{w}^Z(\boldsymbol{x})$ be the decision $\hat{w}_{\epsilon,0}(\boldsymbol{x})$ as defined in (9) when given training a dataset $Z$. Under the stability Assumption B.1 and Assumption B.2, for dataset $Z = \{(\boldsymbol{x}^i, \boldsymbol{\nu}^n)\}_{n=1}^N$ and perturbed dataset $\tilde{Z} = \{(\boldsymbol{x}^i, \boldsymbol{\nu}^n + \delta^n)\}_{n=1}^N$ with any, including potentially adversary, noise $\|\delta^n\| \leq \alpha$,*

$$\mathbb{E}_{\boldsymbol{x}} \left\| \hat{w}^Z(\boldsymbol{x}) - \hat{w}^{\tilde{Z}}(\boldsymbol{x}) \right\|_2 \leq c_0 \lambda(\alpha) \tag{43}$$

*for some constant $c_0$. Additionally, $\lambda(\alpha)$ is an increasing function of the noise level $\alpha$ and $\lambda(0) = 0$ defined explicitly by*

$$\lambda(\alpha) = \max_k \mathbb{P}_x \left( w^*(\boldsymbol{\nu}_x) \in H_k^{\epsilon + \alpha L} \setminus H_k^{\epsilon - \alpha L} \right). \tag{44}$$

*Moreover, this change is concentrated around its mean:*

$$\mathbb{P}_{\boldsymbol{x}} \left( \left\| \hat{w}^Z(\boldsymbol{x}) - \hat{w}^{\tilde{Z}}(\boldsymbol{x}) \right\| - \mathbb{E} \left\| \hat{w}^Z(\boldsymbol{x}) - \hat{w}^{\tilde{Z}}(\boldsymbol{x}) \right\| \geq t \right) \leq \exp \left( \frac{-2t^2}{N} \right). \tag{45}$$

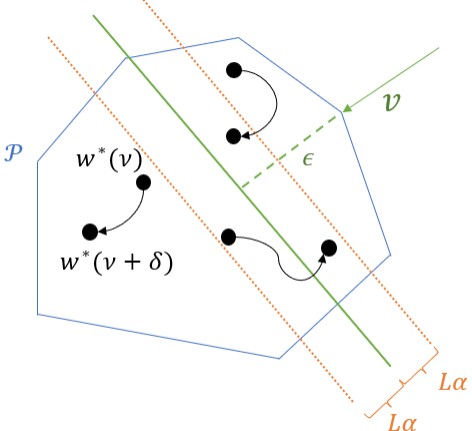

Figure 4: Any point $w^*(\nu)$ belonging to the cap of size $\epsilon - L\alpha$ or outside of the cap of size $\epsilon + L\alpha$ will never change its label after noise is added. In the figure, this is any point outside of the strip defined by the dotted lines. Otherwise, points within the strip may change their label.

**Proof:** First, we show that the labelling $p_k^n$ as defined in 8 does not change for a constant fraction of the data even under noisy perturbations. In particular, note that for any $v^n$ such that $w^*(\nu^n) \in H_k^{\epsilon - L \cdot \alpha}$ we also have $w^*(\nu^n + \delta^n) \in H_k^\epsilon$. Indeed, if $g_{\nu^k}(w^*(\nu^n) - g_{\nu^k}(w^*(\nu^k)) \leq \epsilon - L\alpha$, then by the Lipschitz property B.2, we have $g_u(w^*(\nu^n + \delta^n) - g_{\nu^k}(w^*(\nu^k)) \leq \epsilon$. Therefore, the labeling of $w^*(\nu)$ and $w^*(\nu^n + \delta^n)$ would both be equal to 1 for any noise $\|\delta\|^n \leq \alpha$. Similarly, if $w^*(\nu^n) \notin H_k^{\epsilon + \alpha L}$ then also $w^*(\nu^n + \delta^n) \notin H_k^\epsilon$ and so the labels would both be 0. On the other hand, if $w^*(\nu^n) \in H_k^{\epsilon + L\alpha} \setminus H_k^{\epsilon - L\alpha}$ then it is possible for the label to change. See Figure 4. However, this can only be true for at most a constant fraction of the data, independent of the size of the data. Let this fraction be $\lambda(\alpha)$. That is,

$$\lambda(\alpha) = \max_k \mathbb{P}\left(w^*(\nu) \in H_k^{\epsilon + L\alpha} \setminus H_k^{\epsilon - L\alpha}\right) \tag{46}$$

Clearly, this is an increasing function with $\lambda(0) = 0$.

Notice that the learners $\hat{p}_k(x)$ are trained only on these labels. Let $\hat{p}_k^Z(x)$ be the model learned when training with data from $Z$ and similarly let $\hat{p}_k^{\tilde{Z}}(x)$ be the model when trained on labels resulting from $\tilde{Z}$. Note that from the argument above, the number of labels that are different in each case is bounded in expectation by $\lambda(\alpha)$. Now, we bound $\left\|\hat{w}^Z(x) - \hat{w}^{\tilde{Z}}(x)\right\|$. By definition and linearity from Assumption B.2,

$$\hat{w}_{\epsilon,0}(x) = \arg\min_{w \in \mathcal{P}} \frac{1}{N} \sum_{i=1}^N \hat{p}_i(x) \max\{R_{\nu^i}(w), 0\} \tag{47}$$

$$= \arg\min_{w \in \mathcal{P}} \frac{1}{N} \sum_{i=1}^N \hat{p}_i(x) g_{\nu^i}(w) \tag{48}$$

$$= \arg\min_{w \in \mathcal{P}} g_{\frac{1}{N} \sum_{i=1}^N \hat{p}_i(x)\nu^i}(w) \tag{49}$$

$$= w^*\left(\frac{1}{N} \sum_i \hat{p}_i(x)\nu^i\right) \tag{50}$$

Then,

$$\left\| \hat{w}^Z(\boldsymbol{x}) - \hat{w}^{\tilde{Z}}(\boldsymbol{x}) \right\| = \left\| w^* \left( \frac{1}{N} \sum_i \hat{p}_i^Z(\boldsymbol{x}) \boldsymbol{\nu}^i \right) - w^* \left( \frac{1}{N} \sum_i \hat{p}_i^{\tilde{Z}}(\boldsymbol{x}) \boldsymbol{\nu}^i \right) \right\| \tag{51}$$

$$\leq L_2 \left\| \frac{1}{N} \sum_i (\hat{p}_i^Z(\boldsymbol{x}) - \hat{p}_i^{\tilde{Z}}(\boldsymbol{x})) \boldsymbol{\nu}^i \right\| \leq L_2 \frac{1}{N} \sum_i |\hat{p}_i^Z(\boldsymbol{x}) - \hat{p}_i^{\tilde{Z}}(\boldsymbol{x})| \left\| \boldsymbol{\nu}^i \right\| \tag{52}$$

Now, noting that $\left\| \boldsymbol{\nu}^i \right\| \leq 1$ by assumption,

$$\mathbb{E} \left\| \hat{w}^Z(\boldsymbol{x}) - \hat{w}^{\tilde{Z}}(\boldsymbol{x}) \right\| \leq L_2 \frac{1}{N} \sum_i \mathbb{E} |\hat{p}_i^Z(\boldsymbol{x}) - \hat{p}_i^{\tilde{Z}}(\boldsymbol{x})| \tag{53}$$

Given that a $\lambda(\alpha) \cdot N$ labels change in expectation, using lemma A.1 gives us that $\mathbb{E}|\hat{p}_i^Z(\boldsymbol{x}) - \hat{p}_i^{\tilde{Z}}(\boldsymbol{x})| \leq 2\lambda(\alpha)N\beta_{N,i}$. It follows that

$$\mathbb{E} \left\| \hat{w}^Z(\boldsymbol{x}) - \hat{w}^{\tilde{Z}}(\boldsymbol{x}) \right\| \leq 2L_2 \frac{1}{N} \sum_{i=1}^N \lambda(\alpha)N\beta_{N,i} = O(L_2\lambda(\alpha)) \tag{54}$$

Moreover, the final tail bound follows directly from Hoeffding's inequality.

**Lemma B.1** *Given that $j$ labels change in the training data, the difference in prediction for any fixed $j$ is bounded as $\mathbb{E}_{\boldsymbol{x}}|\hat{p}_k^Z(\boldsymbol{x}) - \hat{p}_k^{\tilde{Z}}(\boldsymbol{x})| \leq 2j\beta_{N,k}$ for a $\beta_{N,k}$-stable algorithm used to learn each $\hat{p}_k$.*

We now prove Lemma B.1 by induction. Let $Z^1, \ldots, Z^j$ be a sequence of datasets where $Z^1 = Z$ and $Z^j = \tilde{Z}$ and where there is exactly one label changed from dataset $Z^i$ to $Z^{i+1}$. Then applying the lemma for the case $j = 1$ to each pair of consecutive datasets $Z^i, Z^{i+1}$ will prove the result.

We now prove the result for $j = 1$. Here we have exactly one label that is changed from $Z$ to $\tilde{Z}$. Let $s$ be the index of the datapoint which has its label changed. Consider an intermediate dataset $\hat{Z}$ which is equal to $Z$ but which has datapoint $s$ removed. Note that $\hat{Z}$ is also equal to $\tilde{Z}$ but with datapoint $s$ removed as well. We can now apply the definition of hypothesis stability to find that

$$\mathbb{E}_{\boldsymbol{x}} \left| \hat{p}_k^Z(\boldsymbol{x}) - \hat{p}_k^{\tilde{Z}}(\boldsymbol{x}) \right| \leq \mathbb{E}_{\boldsymbol{x}} \left| \hat{p}_k^Z(\boldsymbol{x}) - \hat{p}_k^{\hat{Z}}(\boldsymbol{x}) \right| + \mathbb{E}_{\boldsymbol{x}} \left| \hat{p}_k^{\hat{Z}}(\boldsymbol{x}) - \hat{p}_k^{\tilde{Z}}(\boldsymbol{x}) \right| \tag{55}$$

$$\leq \beta_{N,k} + \beta_{N,k} = 2\beta_{N,k} \tag{56}$$

which proves the claim for $j = 1$ and hence the lemma.

## C  FORMULATIONS AND EXPERIMENTS

### C.1  INVENTORY STOCK PROBLEM

Recall the objective is given as

$$g_d(w) = c_0 w + c_q w^2 + c_b \max\{d - w, 0\} + \frac{1}{2} q_b \max\{d - w, 0\}^2 + \tag{57}$$

$$c_h \max\{w - d, 0\} + \frac{1}{2} q_h \max\{w - d, 0\}^2 \tag{58}$$

where $d$ is the uncertain demand and $w$ is the allocation decision for a product. We provide some details on the approaches we compare against:

**Minimum-violation objective with Bertsimas & Kallus (2020).**  Here we aim to solve the problem of $\min_{w \in \mathcal{P}} \mathbb{E}\left[\max\{R_{\boldsymbol{\nu}_{\boldsymbol{x}}}(\boldsymbol{w}) - \phi, 0\}\right]$ by the method of Bertsimas & Kallus (2020) by thinking of the objective function not as $g_{\boldsymbol{\nu}}(\boldsymbol{x})$ but rather as $\max\{R_{\boldsymbol{\nu}_{\boldsymbol{x}}}(\boldsymbol{w}) - \phi, 0\}$ directly. The formulation is given as

$$\min_{w \in \mathcal{P}} \sum_{i=1}^N \pi^i(\boldsymbol{x}) \max\{R_{\boldsymbol{\nu}_{\boldsymbol{x}}}(\boldsymbol{w}) - \phi, 0\} \tag{59}$$

where the $\pi^i(\boldsymbol{x})$ are weights are generated by K-nearest neighbors (or other ML models like linear or tree models as described in Bertsimas & Kallus (2020)). However, these methods not generate this weighting based on the optimization problem itself. In contrast our proposed methods generates weights explicitly depending on the optimization task.

**CVaR objective.** We also compare against the CVaR objective, which is very similar to ours. Specifically, the objective is given as $\mathbb{E}[R_{\boldsymbol{\nu}_{\boldsymbol{x}}}(\boldsymbol{w})|R_{\boldsymbol{\nu}_{\boldsymbol{x}}}(\boldsymbol{w}) \geq q_\alpha(R_{\boldsymbol{\nu}_{\boldsymbol{x}}}(\boldsymbol{w}))]$ where $q_\alpha(Z)$ is the $\alpha$-quantile of a random variable $Z$. According to Rockafellar & Uryasev (2002) the problem can be reformulated to have objective

$$\beta + \frac{1}{\epsilon}\max\{R_{\boldsymbol{\nu}_{\boldsymbol{x}}}(\boldsymbol{w}) - \beta, 0\} \tag{60}$$

where $\beta$ is a newly introduced auxiliary variable. Using the method of Bertsimas & Kallus (2020), we solve the following formulation:

$$\min_{\boldsymbol{w}\in\mathcal{P}, \beta\in\mathbb{R}} \sum_{i=1}^{N} \pi^i(\boldsymbol{x})\left(\beta + \frac{1}{\epsilon}\max\{R_{\boldsymbol{\nu}^i}(\boldsymbol{w}) - \beta, 0\}\right) \tag{61}$$

where again $\pi^i(\boldsymbol{x})$ are determined by the $K = 10$ nearest training neighbors of the out-of-sample point $\boldsymbol{x}$.

**Traditional robust optimization.** We also compare against a more traditional robust optimization approach. Specifically, for out-of-sample $\boldsymbol{x}$, find the $K$ nearest neighbors in the data, namely $\mathcal{N}(\boldsymbol{x}, K)$. Unlike Bertsimas & Kallus (2020) which simply minimizes the average cost, we weight the remaining $K$ datapoints adversarially according to a distribution $\pi$ that is within a Kullback-Leibler divergence distance of at most $r$ from uniform distribution assigning weights $1/K$ to each datapoint (so $D_{KL}(\pi||1/K) \leq r$). The formulation is

$$\arg\min_{w\in\mathcal{P}} \max_{\pi:D_{KL}(\pi||1/K)\leq r} \sum_{i\in\mathcal{N}(\boldsymbol{x},K)} \pi^i g_{\boldsymbol{\nu}^i}(w). \tag{62}$$

This Kullback-Leibler uncertainty set was introduced in Ben-Tal et al. (2013) in the feature-less setting. We use the methods in this paper to solve the proposed method in (62).

## C.2 LOAD FORECASTING

$$g_{\boldsymbol{d}}(\boldsymbol{w}) = \sum_{i=1}^{24} \gamma_s \max\{\boldsymbol{d}_i - \boldsymbol{w}_i, 0\} + \gamma_e \max\{\boldsymbol{w}_i - \boldsymbol{d}_i, 0\} + \frac{1}{2}(\boldsymbol{w}_i - \boldsymbol{d}_i)^2 \tag{63}$$

with additional ramping constraints $\boldsymbol{w}_{i+1} - \boldsymbol{w}_i| \leq r$, limiting the change in electricity generation from one hour to the next. The optimization problem, given a forecast of $\boldsymbol{\nu}$ is given by

$$
\begin{aligned}
w^*(\boldsymbol{d}) = \quad &\arg\min_{\boldsymbol{w}} \quad \sum_{i=1}^{24} \gamma_s \max\{\boldsymbol{d}_i - \boldsymbol{w}_i, 0\} + \gamma_e \max\{\boldsymbol{w}_i - \boldsymbol{d}_i, 0\} + \frac{1}{2}(\boldsymbol{w}_i - \boldsymbol{d}_i)^2 \\
&\text{subject to} \quad |\boldsymbol{w}_{i+1} - \boldsymbol{w}_i| \leq r, \quad i = 1, \dots, 23 \\
&\qquad\qquad \boldsymbol{w} \geq 0
\end{aligned}
\tag{64}
$$

We can rewrite the terms $\max\{\boldsymbol{d}_i - \boldsymbol{w}_i, 0\}, \max\{\boldsymbol{w}_i - \boldsymbol{d}_i, 0\}$ by variables $\boldsymbol{h}_i$ and $\boldsymbol{b}_i$ with constraints $\boldsymbol{h}_i \geq \boldsymbol{d}_i - \boldsymbol{w}_i, \boldsymbol{h}_i \geq 0$ and similar for $\boldsymbol{b}_i$. Since the objective is to minimize the overall cost, the variable $\boldsymbol{h}_i$ will take the minimum possible value while satisfying the constraints. So, it would equal the maximum of 0 and $\boldsymbol{d}_i - \boldsymbol{w}_i$. The problem can be reformulated as

$$
\begin{aligned}
w^*(\boldsymbol{d}) = \quad &\arg\min_{\boldsymbol{w}} \quad \sum_{i=1}^{24} \gamma_s \boldsymbol{h}_i + \gamma_e \boldsymbol{b}_i + \frac{1}{2}(\boldsymbol{w}_i - \boldsymbol{d}_i)^2 \\
&\text{subject to} \quad |\boldsymbol{w}_{i+1} - \boldsymbol{w}_i| \leq r, \quad i = 1, \dots, 23 \\
&\qquad\qquad \boldsymbol{h}_i \geq \boldsymbol{d}_i - \boldsymbol{w}_i, \quad i = 1, \dots, 24 \\
&\qquad\qquad \boldsymbol{b}_i \geq \boldsymbol{w}_i - \boldsymbol{d}_i, \quad i = 1, \dots, 24 \\
&\qquad\qquad \boldsymbol{w}, \boldsymbol{h}, \boldsymbol{b} \geq 0
\end{aligned}
\tag{65}
$$

The method of Donti et al. (2017) views the problem as a stochastic optimization problem where they predict a distribution for the demand $\boldsymbol{d}$ and solves the corresponding stochastic problem:

$$
\begin{aligned}
\min_{\boldsymbol{w}} \quad & \sum_{i=1}^{24} \mathbb{E}_{\boldsymbol{d}\sim p(\boldsymbol{d}|\boldsymbol{x})}\left[\gamma_s \max\{\boldsymbol{d}_i - \boldsymbol{w}_i, 0\} + \gamma_e \max\{\boldsymbol{w}_i - \boldsymbol{d}_i, 0\} + \tfrac{1}{2}(\boldsymbol{w}_i - \boldsymbol{d}_i)^2\right] \\
\text{subject to} \quad & |\boldsymbol{w}_{i+1} - \boldsymbol{w}_i| \le r, \quad i = 1, \ldots, 23 \\
& \boldsymbol{w} \ge 0
\end{aligned}
$$

(66)

where $p(\boldsymbol{d}|\boldsymbol{x})$ is the prediction made (end-to-end) of the demand distribution. The paper predicts the mean and variance of the distribution, then assume it follows the corresponding gaussian distribution. They solve the stochastic problem by sequential quadratic programming (SQP) to iteratively approximate the problem. This is computationally much more expensive to do than the simple deterministic quadratic problem one must solve using the discretization method proposed.

### C.3 Portfolio Optimization

We are given $d$ investment options whose random returns are denoted by $r_i, i = 1, \ldots, d$. Let $D$ be the joint distribution of the $r_i$. As data, we use historical stock prices of 8 companies starting from January 2021, until August 2023. Denote by $r^{(1)}, \ldots, r^{(n)}$ the returns observed on the train data

We wish to make decisions $x_i$ for the fraction of a portfolio to invest in option $i$. In this robust setting, we wish to make a decision that maximizes the probability that the return is above a threshold $t$. In particular, take decision given by

$$
\begin{aligned}
\max_x \quad & \mathbb{P}_{r\sim D}\left(r^T x \ge t\right) \\
\text{subject to} \quad & \sum_{i=1}^{d} x_i = 1 \\
& x_i \ge 0
\end{aligned}
$$

(67)

which we solve approximately by minimizing the expected violation across the data:

$$
\begin{aligned}
\min_{x,j} \quad & \sum_{j=1}^{n} q_j \\
\text{subject to} \quad & q_j \ge (r^{(j)})^T x - t \\
& \sum_{i=1}^{d} x_i = 1 \\
& x \ge 0, q \ge 0
\end{aligned}
$$

(68)

**Budget Uncertainty:** We additionally compare against existing robust optimization approaches. constructs an budget uncertainty set. They solve the problem

$$
\begin{aligned}
\max_x \min_{r\in\mathcal{U}^B} \quad & \sum_{i=1}^{d}(\mu_i + \sigma_i r_i)x_i \\
\text{subject to} \quad & \sum_{i=1}^{d} x_i = 1 \\
& x_i \ge 0
\end{aligned}
$$

(69)

where the uncertainty set is given by

$$
\mathcal{U}^B = \{r : \|r\|_\infty \le 1, \|r\|_1 \le \Gamma\}
$$

(70)

and $\Gamma$ is the co-called budget of uncertainty which needs to be tuned.

**Data driven, KL-divergence and variation distance:** Furthermore, we also compare against a data-driven method, as in . Here, one treats the dataset as an empirical distribution and construct an uncertainty set of all probability distributions (on the same support) that has KL-divergence at most some $\epsilon$. In particular, let $\hat{\pi}_j = 1/n, \forall j = 1, \ldots, n$ be the uniform empirical distribution for the $n$ datapoints. Then, define $\mathcal{U}^{KL}$ as the set of all probability $\pi$ whose KL-divergence is at most $\epsilon$ from $\hat{\pi}$:

$$
\mathcal{U}^{KL} = \{\pi : \quad \pi \ge 0, \quad \sum_{j=1}^{n} \pi_j = 1, \quad \sum_{j=1}^{n} \pi_j \log(\pi_j/\hat{\pi}_j) \le \epsilon\}
$$

(71)

Then, the robust decision is given by solving the following:

$$
\begin{aligned}
\max_x \min_{\pi\in\mathcal{U}^{KL}} \quad & \sum_{j=1}^{n} \pi_j \cdot (r^{(j)})^T x \\
\text{subject to} \quad & \sum_{i=1}^{d} x_i = 1 \\
& x_i \ge 0
\end{aligned}
$$

(72)

However, this is a computationally expensive problem as it requires exponential cone constraints. We see this in the increased running time required to solve the problem. Therefore, we also compare with the same data-driven approach but using a simpler uncertainty set. In, particular consider the set of probability distributions $\pi$ that within a 1-norm distance of $\epsilon$ from the empirical $\hat{\pi}_j = 1/n$. We denote this the variation distance uncertainty set $\mathcal{U}^{VD}$:

$$\mathcal{U}^{VD} = \{\pi : \quad \pi \geq 0, \quad \sum_{j=1}^{n} \pi_j = 1, \quad \sum_{j=1}^{n} |\pi_j - \hat{\pi}_j| \leq \epsilon\} \tag{73}$$

### C.4 INVENTORY ALLOCATION WITH NO FEATURES

We now consider an inventory allocation problem with $K$ products from a store and must decide on a number $x_k$ of items of type $k, k = 1, \ldots, K$ to order. Let $d^{(1)}, \ldots, d^{(M)}$ denote the demand vectors on each of the $M$ days of data. $d_k^{(m)}$ denotes the demand of product $k$ on day $m$. We assume a uniform holding cost of $H$ for each unit of product that is not sold and a lost opportunity cost $C$ for each unit of demand is left unmet. The cost of allocating $x$ units when the true demand is $d$ is given by

$$\sum_{k=1}^{K} C \cdot \max\{d_k - x_k, 0\} + H \cdot \max\{x_k - d_k, 0\}. \tag{74}$$

Furthermore, there is a capacity $S$ on the total supply of products allowed in the store. So, $\sum_{k=1}^{K} x_k \leq S$. We wish to make a decision $x$ which maximizes the probability that its cost is below a certain threshold $t$ with demand $d$ coming from some unknown distribution $D$:

$$\begin{array}{ll} \max_x & \mathbb{P}_{d \sim D} \left( \sum_{k=1}^{K} C \cdot \max\{d_k - x_k, 0\} + H \cdot \max\{x_k - d_k, 0\} \leq t \right) \\ \text{subject to} & \sum_{k=1}^{K} x_k \leq S \\ & x_i \geq 0 \end{array} \tag{75}$$

Using the framework of minimizing the expected violation, we reformulate the problem as

$$\begin{array}{ll} \max_{x, y_h, y_c, q} & \sum_{m=1}^{M} q_m \\ \text{subject to} & q_m \geq \sum_{k=1}^{K} H \cdot h_k^m + C \cdot c_k^m - t \\ & h_k^m \geq x_k - d_k^{(m)}, \quad h_k^m \geq 0 \\ & c_k^m \geq d_k^{(m)} - x_k, \quad c_k^m \geq 0 \\ & \sum_{k=1}^{K} x_k \leq S, \quad x_i \geq 0 \end{array} \tag{76}$$

where we introduce variables $h_k^m$ and $c_k^m$ to indicate the number of units that contribute to holding and opportunity costs, respectively.

**Adaptive robust optimization:** We also compare against an adaptive robust optimization approach. Note that this is inherently a two-stage problem: after a demand realization is made, one must fulfil the demand. So, $x_k$ are the first stage allocation decision, and $h_k(d), c_k(d)$ are the second stage variables that describe the fulfilment of demand $d$. Let $\mathcal{U}$ denote the uncertainty set for the uncertain demand $d$. Then, the adaptive robust optimization problem is given as

$$\begin{array}{ll} \min_x \max_{d \in \mathcal{U}} & \sum_{k=1}^{K} H \cdot h_k(d) + C \cdot c_k(d) \\ \text{subject to} & h_k(d) \geq x_k - d_k, \quad h_k(d) \geq 0 \\ & c_k(d) \geq d_k - d_k, \quad h_k(d) \geq 0 \\ & \sum_{i=1}^{d} x_i = 1 \\ & x_i \geq 0 \end{array} \tag{77}$$

However, as written this is an infinite dimensional problem, requiring a different variable $h_k(d)$ for each $d \in \mathcal{U}$. As is commonly done, we use linear decision rules for the second stage variables. Hence, $h_k(d), c_k(d)$ are now linear functions of $d$:

$$h_k(d) = h_k^0 + \sum_{j=1}^{K} h_k^j \cdot d_j \tag{78}$$

$$c_k(d) = c_k^0 + \sum_{j=1}^{K} c_k^j \cdot d_j \tag{79}$$

for new variables $h_k^0, \ldots, h_k^K, c_k^0, \ldots, c_k^K, k = 1, \ldots, K$ to define the linear functions of $d$. As an uncertainty set, we use again a budget uncertainty set:

$$\mathcal{U} = \left\{ d : \sum_{k=1}^{K} \frac{d_k - \mu_k}{\sigma_k} \leq \Gamma \right\} \tag{80}$$

where $\mu_k, \sigma_k$ are the empirical mean and variance of the demand of the $k^{th}$ product. Again, $\Gamma$ is the budget of uncertainty that must be tuned.

**Data driven, KL-divergence and variation distance.** As in the previous section, we consider the data-driven method, robust against all empirical distributions that are within a fixed distance from the observed empirical distribution. Again, let $\hat{\pi}_m$ denote the uniform distribution over the data $d^{(1)}, \ldots, d^{(M)}$. We consider the two uncertainty sets of KL-diveregence $\mathcal{U}^{KL}$ and Variation Distance $\mathcal{U}^{VD}$ as in (71) and (73). The problem can then be formulated as

$$
\begin{aligned}
\min_{x, y_h, y_c} \max_{\pi \in \mathcal{U}^{KL/VD}} \quad & \sum_{m=1}^{M} \pi_m \cdot \left( \sum_{k=1}^{K} H \cdot h_k^m + C \cdot c_k^m \right) \\
\text{subject to} \quad & h_k^m \geq x_k - d_k^{(m)}, \quad h_k^m \geq 0 \\
& c_k^m \geq d_k^{(m)} - x_k, \quad c_k^m \geq 0 \\
& \sum_{k=1}^{K} x_k \leq S, \quad x_i \geq 0
\end{aligned}
\tag{81}
$$

**Results:** For each decision made by all four approaches described above we calculate all costs on test data and report the worst case costs incurred. In figures 5,6 we see the costs of the worst 5% examples for each decision at various different supply capacities. We see that overall our proposed approach is the minimum of the other approaches at almost every quantile. Furthermore, as the capacity increases, and the more freedom we have in the decisions we can make, the larger the gaps become. At the very highest quantiles (99%-100%), we see our approach performing up to 15-20% better than the other approaches. Moreover, in the 95%-97% quantiles, our approach and the adaptive approach have a cost lower by up to 18% compared to the KL and VD approaches when the capacity is 800. At a capacity of 1100, they have a cost up to 25% lower.

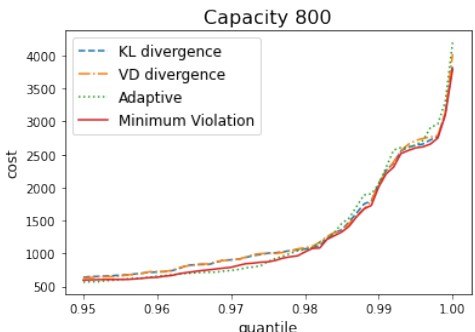

Figure 5: Worst-case costs at 800 capacity.

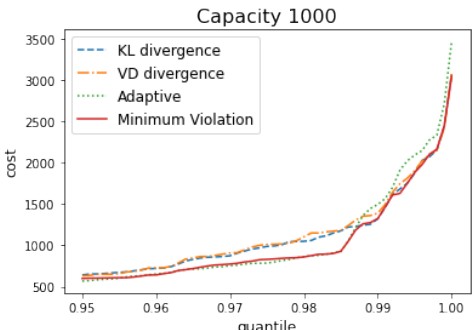

Figure 6: Worst-case costs at 1000 capacity.

## D  IMPLEMENTATION AND TRACTABILITY

We now discuss some details of implementation and improving the running time of the method. There are two parts to the approach, first to learn the labelling $p_k^n$ and second to solve the optimization problem to determine $\hat{w}_{\epsilon, \phi}(\boldsymbol{x})$.

## E  MAXIMIZING PROBABILITY OF LOW REGRET

Finally, we have a brief discussion on the difficulty of the original robust problem proposed in 4:

$$\max_{\boldsymbol{w} \in \mathcal{P}} \mathbb{P}_{\boldsymbol{\nu}_{\boldsymbol{x}} \sim D_{\boldsymbol{x}}} \left( R_{\boldsymbol{\nu}_{\boldsymbol{x}}}(\boldsymbol{w}) \leq \phi \right). \tag{82}$$

In general, if the distribution $D_{\boldsymbol{x}}$ is discrete, having point masses $\boldsymbol{\nu}_{\boldsymbol{x}}^1, \ldots, \boldsymbol{\nu}_{\boldsymbol{x}}^k$ with probabilities $p_1, \ldots, p_k$, the problem can be formulated as an mixed integer optimization problem:

$$\max_{\boldsymbol{w}, \boldsymbol{q}} \quad \sum_{j=1}^{k} p_j \cdot \boldsymbol{q}_j$$

$$\text{subject to} \quad R_{\boldsymbol{\nu}_{\boldsymbol{x}}^j}(\boldsymbol{w}) \leq \epsilon + M(1 - \boldsymbol{q}_j) \tag{83}$$

$$\boldsymbol{w} \in \mathcal{P}$$
$$\boldsymbol{q} \in \{0, 1\}^k$$

If $R_{\boldsymbol{\nu}_{\boldsymbol{x}}^j}(\boldsymbol{w}) > \epsilon$, the constraint forces that $\boldsymbol{q}_j = 0$. Therefore, $\boldsymbol{q}_j$ is a binary variable that reflects whether $R_{\boldsymbol{\nu}_{\boldsymbol{x}}^j}(\boldsymbol{w}) \leq \epsilon$ or not. If indeed the regret is less than $\epsilon$, then $\boldsymbol{q}_j$ can take either the value 0 or 1. But the objective is to maximize the weighted sum of the $\boldsymbol{q}_j$, therefore at optimality, it will take the value of 1 whenever the regret of $\boldsymbol{w}$ with respect to $\boldsymbol{\nu}_{\boldsymbol{x}}^j$ is at most $\epsilon$.

