# OpenReview forum: "A Discretization Framework for Robust Contextual Stochastic Optimization"
_ICLR.cc/2024/Conference — ICLR 2024 poster_

### Official Review · Reviewer_9FbH · 2023-10-26

**Soundness:** 4 excellent
**Presentation:** 3 good
**Contribution:** 3 good
**Rating:** 8
**Confidence:** 3

**Summary:**

The authors propose a new approach to contextual stochastic optimization. This approach first divides the feasible region into subregions, and then builds a prediction model that predicts the probability that the optimal solution for a given context will fall in a given subregion. These predicted probabilities are incorporated into an optimization problem that can be used to select a decision for a given context. The authors provide asymptotic bounds regarding the generalization error of the method as well as a type of stability of the method. The authors provide computational results demonstrating the results of the method for a newvendor problem and a generator scheduling problem.

**Strengths:**

The author's approach to contextual stochastic optimization problems is overall quite novel. The paper is well-organized and clearly written. Contextual optimization problems are an important topic of recent interest, so novel methods for this class of problems are a welcome contribution.

**Weaknesses:**

There is a significant issue with the framing of the paper. In particular, the authors frame their contribution as an approach for robust contextual stochastic optimization problems. However, I do not think that this an accurate presentation. Consider the central optimization problem (equation 5, page 4):
$$\min_{w \in \mathcal{P}} \mathbb{E}_{\nu_x \sim D_x}[\max\\{R\_{\nu\_x}(w)-\phi,0\\}]$$
Note that this problem can be rephrased as the problem of minimizing the expectation of a function $f\_{\nu\_x}$, specifically:
$$f\_{\nu\_x}(w) = \max\\{R\_{\nu_x}(w)-\phi,0\\}$$
So, this problem is just a special type of a contextual stochastic optimization problem, and techniques for contextual stochastic optimization problems, such as that in Bertsimas and Kallus (2020) or Donti et al. (2017), could be applied here. Note that robust, distributionally robust, or risk-averse optimization problems cannot typically be easily reduced to a standard stochastic problems.

The authors do not provide a proof of Lemma A.1 in the appendix, which seems like a fairly critical step in the proof of Theorem 5.1. In addition, I suspect that Lemma A.1 is not actually true. In particular, the Hypothesis Stability provides a convergence rate $\beta_N$ on the change in predicted probability by omitting a point for a single binary classification problem. However, it seems to me for this lemma to work, the authors would need a uniform convergence rate $\beta_N$ that applies to all $N$ binary classification problems (note here that the number of binary classification problems appearing in the authors' method increases with sample size). This seems to me to be a much stronger property.

The computational experiments compare the authors' methods to methods that solve a completely different problem. In particular, three of the four comparison methods seem to be designed to solve the problem $\min\_{w}\mathbb{E}\_{\nu\_x \sim D\_x}[R\_{\nu\_x}(w)]$ , while the remaining method (KNN+KL Robust) is designed to solve a robust variant of this problem that is not clearly connected to the problem that the authors are solving. Notably, the authors claim that the method of Bertsimas and Kallus (2020) does not apply to this problem. However, this method can be applied as follows. First, fit a prediction model that predicts the parameter $\nu$ from the context $x$. Then, given some fixed context $x$, derive weights $\hat{p}\_n(x)$ for each training points $\{x^N,\nu^N\}$ (as described in Bertsimas and Kallus 2020).  Then, solve the problem:
$$\min_{w \in \mathcal{P}} \sum_{n=1}^N \hat{p}\_n(x)\max\\{R\_{\nu^n}(w)-\phi,0\\}$$

Similarly, the method of Donti et al. (2017), the "policy optimizer method" and the "MLE method" could also be applied to the problem $\min_{w \in \mathcal{P}} \mathbb{E}_{\nu_x \sim D_x}[\max\\{R\_{\nu\_x}(w)-\phi,0\\}]$, rather than to the problem of $\min\_{w}\mathbb{E}\_{\nu\_x \sim D\_x}[R\_{\nu\_x}(w)]$ by making relatively straightforward modifications to these methods.

Minor comments:

The authors state that $\mathbb{E}\_{\nu\_x \sim D\_x}[R\_{\nu\_x}(w)-\phi \mid R\_{\nu\_x}(w) \geq \phi]$ is the CVar (discussion following equation 6 in page 4). However, this is not accurate. The CVaR in this setting would be given by something like: $ \mathbb{E}_{\nu_x \sim D_x}[R\_{\nu\_x}(w) \mid R\_{\nu\_x}(w) \geq q\_{\alpha}(R\_{\nu\_x}(w))]$ where $q\_{\alpha}(Z)$ gives the $\alpha$ quantile of the variable $Z$. Minimizing the latter is not equivalent to minimizing the former.

The authors frequently refer to expressions involving $w^*$ that are only well-defined if the problem has a unique solution. This could easily be fixed, for example by stating that $w^*$ is an oracle that chooses one solution.

**Questions:**

Why did you choose this particular form of objective function, rather than using more established notions of risk, such as CVaR?

---

> ### Author Response · Authors · 2023-11-15
>
> Thank you  for your comments. We have done our best to address them in the revised version of the paper.
> In terms of terminology, indeed our approach is not solving a form of the traditional robust optimization problem. However, the end result is still to make decisions which are robust to uncertainty.
>
> We apologize for missing the proof of Lemma A.1 (now Lemma B.2). We have included it in the revised version of the paper. For your question, we assume the hypothesis stability holds true for each $\hat{p}_k(\cdot)$ separately. We have revised the proof of the theorem to be clearer and easier to follow as well. Please let us know if you have any questions.
>
> We also agree we should compare against other methods of solving the minimum-violation objective we propose. We have added the approach of solving this using the method of Bertsimas and Kallus (2020) as you suggested.
>
> For the ML methods such as Donti et al. (2017) and the ``policy optimizer method'' we noticed this approach unfortunately does not work well. We noticed the following pattern. Whenever the objective for a datapoint is below the threshold $\phi$, the gradient is zero and the weight update does not take these datapoints into account. Rather the update minimizes cost for any datapoints that previously resulted in cost above $\phi$. However, this can result in the datapoints which had cost below $\phi$ before the gradient update step, to now have a cost above $\phi$ after the update. Ultimately, this only creates a mask for which datapoints are updated at each step. In the experiments, we saw that $\phi$ had very little effect (to no effect) on changing the results at any of the quantiles or the mean. We are happy to add some discussion of this and experimental results in the appendix in the following days.
>
> We still hope you find it valuable to compare against other objectives such as minimizing the mean and the robust optimization approach. From the comments of another reviewer, we also now included comparisons with CVaR. The goal is to validate the choice itself of using this minimum-violation objective.
>
> We made this choice of objective because it very naturally aligns with the construction of the sets $H^\epsilon_k$. We have added a discussion on this in the revised version of the paper at the end of Section 3. It is perhaps less clear whether there is such a clean connection when considering an approach such as CVaR instead. Nevertheless, we also agree that this is an interesting question and a possible extension to the paper (thank you!).

---

> > ### Comment · Reviewer_9FbH · 2023-11-20
> >
> > Thank you for the changes to the paper. I believe that the paper is much stronger now, and would revise my score to an 8 (accept).
> >
> > A couple of relatively minor follow-up comments:
> >
> > Regarding the stability results, I appreciate the corrections and changes made to the proof. However, I would disagree with the statement, "we assume the hypothesis stability holds true for each $\hat{p}_k(\cdot)$ separately". The hypothesis stability holding for each problem separately would typically mean that for each $k$ there exists a rate $\\{\beta\_{n,k}\\}\_{n=1}^\infty$ satisfying the given conditions. The authors instead assume that there exists a rate $\\{\beta\_{n}\\}\_{n=1}^\infty$ that applies uniformly for all problems. I would further note that Assumption B.1 is ambiguous in this regard; it's unclear whether the authors are referring to the former situation or the latter. This should be clarified.
> >
> > I think that the discussion following equation (6) still gives the impression that the authors' expression is equivalent to optimizing CVaR, which is still not the case. For a fixed values of  $w$, say $w_1$ and $w_2$, the resulting expression is the CVaR for some quantiles $\alpha_1$ and $\alpha_2$, but the quantiles $\alpha_1$ and $\alpha_2$ are different. If the authors want to compare their expression to CVaR, they should make this more clear.

---

> > > ### Author Response · Authors · 2023-11-21
> > >
> > > Thank you very much for the update.
> > >
> > > And thank you for the follow-up comments. Yes, the rate can certainly be different for each $k$ and we will clarify this in the assumption, as well as the proof.
> > >
> > > That's a good point for the comparison with CVaR, we can add this in the discussion in the paper to make the distinction clearer.
> > >
> > > We have updated the paper with these suggestions.

---

### Official Review · Reviewer_VbmN · 2023-10-29

**Soundness:** 2 fair
**Presentation:** 3 good
**Contribution:** 2 fair
**Rating:** 6
**Confidence:** 3

**Summary:**

The paper proposed a new data-driven approach for the contextual optimization problem. It also provides a new loss function to allow users to balance the robustness and the expected performance. To illustrate the performance, the authors provide a theoretical analysis of the regret and algorithm stability as well as the numerical results for many applications.

**Strengths:**

1. The idea is new and interesting, which is to discretize the decision set to do the robust optimization looks interesting.

2. The new loss function can balance the worst-case regret and the expected regret.

2. the numerical performance looks good.

3. the paper has theories corresponding to the algorithm.

**Weaknesses:**

1. My main concern is the efficiency of the given algorithm. For me, it might be impossible to efficiently learn $\hat{p}$ in Step 3 of the algorithm unless the size of the training sample is exponentially large as the dimension of the feasible set. Please see question 1 for more details.

2. It seems that the feasible set must be fixed for different contextual information.

3. It would be better to have more discussion about the choice of $\phi$ in the loss function.

Some minor points:

4. In addition, It looks like the bound in Theorem 4.1 is established for any fixed decision $w$. It would be better to have a uniform generalization bound instead since the output of the algorithm is not a fixed action.

5. Assumption 5.2 looks strong for many optimization problems. For example, linear programming does not satisfy (14).

**Questions:**

1. Corresponding to Part 1 in the weaknesses, I wonder whether one can learn the probability efficiently. Or, equivalently, I wonder whether the authors could elaborate more on the sample complexity for learning and reducing $\Epsilon$ in (11) for Theorem 1. Please correct me if I am wrong. For me, I am not sure whether it is possible to learn $\Epsilon$ efficiently, especially when the dimension of the feasible set is large. In other words, one may require the size of training data to be exponentially large as the dimension of the feasible set to have a low $\Epsilon$. Particularly, based on the algorithm, H_k^{\epsilon} may only cover a small region, the volume of which can decrease exponentially when the dimension is high. Then, if the size of the training samples is only polynomially large as the dimension, it is very likely that $H_k$'s are disjointed or rarely have non-empty intersections. In this case, the multi-label data set for each training sample might be a vector with only one non-zero entry.

2. Can the algorithm still be useful when the feasible region also depends on the contextual data?

3. In practice, I wonder how one should choose $\phi$ and $\epsilon$.

4. Corresponding to part 4 in the weaknesses, I wonder whether the authors would like to change Theorem 4.1 to a uniform generalization bound.

---

> ### Author Response · Authors · 2023-11-15
>
> Thank you for the thoughtful questions and suggestions. We agree especially that the learnability of the problem  and choice of $\epsilon$ are very important. To address these issues, we have added a discussion on the effect of $\epsilon$ on each term of the bound. Indeed, you are correct, in the worst case one would need an exponential amount of data for small $\epsilon$. Nevertheless, instead, one can increase the value of $\epsilon$ so that the number of positive labels increases. Of course, in the worst case  this may require the choice of $\epsilon$ to be a constant fraction of the diameter of the feasible region.
>
> Primarily, the choice of $\epsilon$ will depend on the distribution of  $w^*(\nu)$. There are two factors that can reduce the dimensionality of this. (1) The $w^*({\nu})$ often belong to a lower-dimensional subspace of $\mathbb{R}^d$ since they must satisfy constraints. For example, if $\mathcal{P}$ is the set of {$ \{ w : Aw = b, w \geq 0 \}$} then the effective dimension of the problem is equal to the dimension of the null space of $A$. For instance, if $Aw = b$ defines a 1-dimensional line, we can view the problem as 1-dimensional regardless of the ambient dimension of the decisions $w \in \mathbb{R}^d$.  (2) the distribution further depends on the data itself. The distribution of ${\nu}$, and as a result of $w^*({\nu})$, does not pathologically cover the entire space when presented with real-world data.
>
> This dimensionality issue is not unique to our approach. The same problem is encountered for example, for the methods of Bertsimas and Kallus (2020) and more generally for methods such as KNN. That being said, we agree with you that it is important to address this issue in the paper.
>
> Finally, that is an interesting extension to consider if the feasible region also depends on the contextual data and uncertainty. That is something we have been looking into and can add in the future.

---

> > ### Comment · Reviewer_VbmN · 2023-11-22
> >
> > Thank you for your clarification. I would like to increase my rating to 6!

---

> > > ### Author Response · Authors · 2023-11-22
> > >
> > > Thank you for the update! And thank you again for the review and thoughtful comments, and in improving the scope and discussion of the theorem. These have all been very helpful in improving the paper.

---

### Official Review · Reviewer_67NA · 2023-10-30

**Soundness:** 3 good
**Presentation:** 2 fair
**Contribution:** 3 good
**Rating:** 8
**Confidence:** 4

**Summary:**

This paper proposes and studies a novel robust optimization framework that is based on discretization of the domain with respect to the data set. The framework proceeds as follows. For each data sample about the problem uncertainty, it constructs the set of near-optimal solutions to the optimization problem whose parameters are specified by the data sample. These sets basically give rise to the "discretization" of the solution space. Then the framework builds an ML model to learn the probability that the optimal solution to the problem whose parameter is given by a particular data sample is near-optimal with respect to other data samples. Here, the framework is flexible in terms of choosing which ML models to use, e.g., k-nearest neighbor. The paper provides some theoretical performance guarantees of the proposed framework and promising numerical results.

However, it seems that the paper could be further improved by investigating further the connections between the proposed framework and the existing data-driven robust optimization methods. Moreover, the paper needs to refine and better present its theoretical performance guarantees. Lastly, the reader would appreciate if the paper provided more intuitions about the proposed framework for robust optimization problems particularly.

**Strengths:**

* The robust optimization framework of solving minimizing $E[[R(w)-\phi ]_+]$ is novel while it is related to many of the existing robust optimization frameworks.
* The data-driven learning-and-optimization method studied in this paper is novel, and the method is flexible in that it can take any ML models to learn the uncertain parameters of the problem.
* The paper provides both theoretical performance analysis and promising numerical results.

**Weaknesses:**

* It is difficult to appreciate the main theoretical performance guarantee on the proposed framework  (given in Theorem 4.1) due to its dependence on the parameters $c$ and $\alpha$. Here, $c$ would increase as the accuracy parameter $\epsilon$ as well as the size of each subset decrease, in which case the bound gets weak. We may try to decrease the parameter $\epsilon$ to make $c$ close to 1, but this would make the mean prediction error $\mathcal{E}$ large. At the same time, no explicit dependence of $c\mathcal{E}$ on $\epsilon$ is studied. Lastly, $\alpha$ could be arbitrarily large when the objective function is not bi-Lipschitz.
* The framework is almost equivalent to minimizing the conditional value-at-risk (CVaR), except that there is an additional probability term being multiplied. The paper would be stronger if it discussed the relationship between the existing CVaR minimization setting and the framework of this paper. No numerical results were done with the CVaR framework.
* There are many typos in the mathematical proofs in the appendix.

**Questions:**

* On page 4, the algorithm is described. Should each data be given by $(x^n, (p_k^n)_{k=1,\ldots, N})$ instead of $k=1,\ldots, n$?
* Typically, the notion of regret is defined for iterative learning processes. Is it possible to extend the current framework to the online setting for which we allow updating the model with a new set of data samples.
* In the introduction, it is stated that the difference between the in-sample cost and the out-of-sample cost decreases on the order of $1/\sqrt{n}$. Perhaps, it would be a better strategy to decompose the current statement of Theorem 4.1 into two where one explains the statement and the other presents the regret bound.
* Could you provide more computational results on a broad set of tolerance levels? For example, 50%, 75%, 90%, 95% quantiles.

---

> ### Author Response · Authors · 2023-11-15
>
> Thank you very much for your comments and review. Taking your comments into account, we hope that the new section we added regarding the impact of $\epsilon$ upon each of the terms $c \cdot \mathcal{E}$ and $c \cdot OBJ({w})$ addresses your questions and concerns. Ultimately, both $c \cdot \mathcal{E}$ and $c \cdot OBJ({w})$
> are always bounded by a constant for any $\epsilon$. The only term left is the generalization term $c \cdot \sqrt{\log(1 / \delta) / N}$. From here we give a procedure on how to  choose $\epsilon$ based on the amount of data available. This gives us a guide to understand the magnitude needed for $\epsilon$, but ultimately something like cross-validation can be used to make a final choice for $\epsilon$ in practice.
> Please see the discussion following the theorem statement that contains more details on this.
>
> We have also added experimental results for a broader range of quantiles, as well as comparing our approach to CVaR as you have suggested. We notice that the CVaR approach produces decisions that change discretely as the robustness parameter changes (the quantile being targeted), whereas the minimum-violation objective produces continuously changing decisions. This continuity might be explained, as you pointed out, by the additional probability term being multiplied. From the computational experiments we also see that our proposed approach outperforms CVaR (and all approaches we considered) by the following metric: for the same average cost, the cost at the $q^{th}$ quantile is generally lower for our approach. This holds in particular for higher quantiles. Please see the plots in Figure 2 that provides many comparisons.
>
> This is an interesting question about considering the online learning setting. In our case, we decided to use regret because it essentially normalized the objective function ---
> so that when applying the optimal decision we obtain a regret of $0$. However, if one only uses the objective function (and not consider regret), the optimal decision can have a different cost for every realization of ${\nu}$. Of course, the method we propose applies just as easily when considering only the objective, and not regret.

---

> > ### Comment · Reviewer_67NA · 2023-11-22
> >
> > Thank you for improving the paper. I appreciate the clarifications you added after Theorem 4.1, which better explain the performance of the proposed framework. In terms of numerical experiments, I was a bit concerned about the computational impact of the method on lower quantile values, as the initial set of results were only about 1% worst case scenarios. Given that the method works well with even 85% quantile scenarios, I am more convinced that the method can be an effective alternative for robust optimization. That said, I have raised my score to 8.

---

> > > ### Author Response · Authors · 2023-11-22
> > >
> > > Thank you very much for the update. And thank you for the thoughtful suggestions to improve the paper, both regarding the discussion for the theorem and in improving the experimental section.

---

### Official Review · Reviewer_XXVR · 2023-10-31

**Soundness:** 2 fair
**Presentation:** 2 fair
**Contribution:** 2 fair
**Rating:** 3
**Confidence:** 3

**Summary:**

This paper explores robust contextual stochastic optimization. In this problem setting, we have a feasible set $\mathcal{P}$ and an objective function $g_{\bm{v}}(\bm{\omega})$, where $\bm{v}$ is a random variable. The distribution of $\bm{v}$ depends on a contextual variable $\bm{x}$, denoted as $\bm{v} \sim \mathcal{D}_{\bm{x}} $, and this distribution is unknown. Unlike other contextual stochastic optimization problems, the primary focus of this paper is the following problem:
$
\min_{\bm{\omega} \in \mathcal{P}} \mathbb{E}_{\bm{v}_{\bm{x}} \sim \mathcal{D}_{\bm{x}}} \left[\max\{R_{\bm{v}_{\bm{x}}}(\omega)-\phi, 0 \}\right]
$
This problem aims to minimize the expected violation, where $\phi$ represents a user-defined cost threshold. Here, $R_{\bm{v}}(\omega) = g_{\bm{v}}(\bm{\omega}) - g_{\bm{v}}(\bm{\omega^*(\bm{v})})$ denotes the regret of decision $\bm{\omega}$ when the objective function parameter is $\bm{v}$.

\section*{Methodology}

Given a historical dataset of observations $\{(\bm{x}^i, v^i)\}_{i=1}^n$, the authors propose to define two key components:
$
H_k^\epsilon = \{\bm{\omega}\in \mathcal{P}: R_{\bm{v}^k}(\omega) \leq \epsilon\},
$
and
$
p^n_k = \begin{cases}
    1, & \text{ if } \bm{\omega}^*(\bm{v}^n) \in H_k^\epsilon
    \\\\
    0, & \text{ otherwise}.
\end{cases}
$

To harness this information, the authors propose training a machine learning model $\hat{p}_k^\epsilon(\bm{x})$ to predict $\mathbb{P}(\bm{\omega}^*(\bm{v}_{\bm{x}}) \in H_k^\epsilon)$. Consequently, the contextual stochastic optimization problem transforms into:
$
\min_{\bm{\omega}} \sum_{k=1}^N \hat{p}_k^\epsilon(\bm{x}) \max\{R_{\bm{v}_{\bm{x}}}(\omega)-\phi, 0\}.
$

\section*{Questions and Comments}

In reviewing this work, several questions and comments have emerged:

\begin{itemize}
    \item First, regarding the definition of $H_k^\epsilon$, it seems unnecessary to have that $\cup_{k=1}^n H_k^\epsilon = \mathcal{P}$. If this is the case, it is unclear how the authors intend to bound the regret when $\bm{x}$ is observed and $\mathbb{P}\left(\bm{\omega}(\bm{v_x}) \in \mathcal{P}\setminus (\cup_{k=1}^n H_k^\epsilon)\right) > 0$.

    \item Additionally, the paper would benefit from a discussion of how the machine learning model $\hat{p}_k^\epsilon(\bm{x})$ is trained based on the provided data. Are the models trained jointly or separately on $(p_k^n)_{k=1, \cdots, n}$? If trained separately, there may be cases where $\sum_{k=1}^N \hat{p}_k^\epsilon(\bm{x}) \neq 1$, making it challenging to interpret $\min_{\bm{\omega}} \sum_{k=1}^N \hat{p}_k^\epsilon(\bm{x}) \max\{R_{\bm{v}_{\bm{x}}}(\omega)-\phi, 0\}$ as an approximation of $\min_{\bm{\omega}} \mathbb{E}_{\bm{v}_{\bm{x}} \sim \mathcal{D}_{\bm{x}}} \left[\max\{R_{\bm{v}_{\bm{x}}}(\omega)-\phi, 0 \}\right]$. In cases of joint training, how do the authors handle models when the possible outcomes $(p_k^n)_{k=1, \cdots, n}$ outnumber the training data size? Regrettably, these details are not addressed in the numerical study neither.

    \item Furthermore, when examining the final problem:
$
    \min_{\bm{\omega}} \sum_{k=1}^n \hat{p}_k^\epsilon(\bm{x}) \max\{R_{\bm{v}_{\bm{x}}}(\omega)-\phi, 0\}
$
    it becomes apparent that, although it is linear in $n$, when $n$ is very large, this problem may become computationally challenging. Do the authors have any insights or strategies for addressing this issue?

    \item Finally, the overall framework resembles a weighted average optimization problem. It would be meaningful to discuss how this framework fundamentally differs from Bertsimas and Kallus (2020) and related literature.

    \item A recommendation is made to move Section 5 entirely to an appendix, allowing more space for a deeper discussion of the framework, algorithm, and implementation details.
\end{itemize}

\section*{Minor Comments}

Some minor comments for consideration:

\begin{itemize}
    \item In Equation (10), $H_\epsilon^k$ should be corrected to $H_k^\epsilon$.

    \item In the beginning of the sixth line in Theorem 4.1, $H_k$ should be corrected to $H_k^\epsilon$.

    \item In the penultimate line of the same theorem, $x \to \bm{x}$ and $\bm{v}_x \to \bm{v_x}$ for consistency.
\end{itemize}

**Strengths:**

Proposed a new framework to solve robust contextual stochastic optimization problem. Please see my comments in Summary for details.

**Weaknesses:**

Not fully clear presentation/discussion of their algorithm. Please see my comments in Summary for details.

**Questions:**

Please see my comments in Summary for details.

---

> ### Author Response · Authors · 2023-11-15
>
> We really appreciate your comments and review. We have done our best to incorporate your suggestions and updated the paper accordingly. We would also like to take the opportunity and answer some of your questions in this response as well.
>
> We agree that it would be very helpful to include a more thorough discussion about how the model is trained, and how the parameters are chosen. In particular, please see the new discussion we added in the revised paper following the theorem about the role of $\epsilon$ on the bound and on learning the  $\hat{p}^{k}({x})$ (the learned probabilities of $w^*({\nu_x}) \in H^\epsilon_k$).
>
> In what follows we try to address some of the questions you raised.
>
> Indeed, the union of $H_k^\epsilon$ need not cover the entire feasible region. This is part of the bound in Theorem 4 and captured in c which depends on $\mathbb{P}(R_{\nu_y}(w^*(\nu_x) \leq \epsilon)$. This is the probability that the objective of a solution for $\nu_x$ is within $\epsilon$ of the optimal solution with respect to some other ${\nu}_{{y}}$.
>
> The values of $\hat{p}_k({x})$ learned need not sum to 1. This is because the sets $H^\epsilon_k$ are not disjoint and do not  necessarily cover the entire feasible region.
> There is a lot of freedom in how the $\hat{p}_k({x})$ can be learned.
> In the first experiment on the inventory problem, each $\hat{p}_k({x})$ comes from a separate knn predictor. For the second experiment on the generator scheduling problem, $\hat{p}_k({x})$ are trained jointly and are the output of a neural network with $N$ outputs (one for each $k =1, \dots, N$).
>
> As also pointed out by reviewer 9FbH, the method indeed has similarities to that of (Berstimas and Kallus (2020)). In particular, one can also view the objective function as some new $f_{{\nu}}({w}) = \max \{ R_{{\nu}}({w}) - \phi, 0 \} $ and directly apply their method on this objective.
>
> To the best of our knowledge, we are the first to propose this new objective function.  More importantly, the significant difference lies in how the weights $\hat{p}_{k}({x})$ are defined. In our case, these are explicitly informed by the optimization problem since $H_k^\epsilon$ are directly dependent on the data and objective function.
>
> From  a computational viewpoint, indeed the problem increases linearly with the amount of data $N$. However, there are three possible approaches to improve the runtime. One is to simply choose a random subset of the data to use to generate sets $H^\epsilon_k$. The second is, as we run the optimization problem for various out-of-sample points ${x}$, to keep track of the ${x}$ and the corresponding solutions $\hat w_{\epsilon, \gamma}(x)$ found.
>  When given a new ${x}'$ we can for example find the nearest neighbor, say ${x}''$ and use $\hat{w}_{\epsilon, \gamma}({x}'')$ as a warm start when solving the optimization problem for ${x}'$. Finally, once calculating all values of $\hat{p}_k({x})$, many of these will have very low value close to zero. One approximation is to simply remove these from the optimization problem. Or alternatively, only keep the largest $k$ values of $\hat{p}_k({x})$ and only optimize with respect to those. We are happy to add more discussion on this in the appendix of the revised paper.

---

### Author Response · Authors · 2023-11-15
**Common Response and Updates**

We would like to thank the reviewers for their comments and suggestions. These have been incredibly helpful in improving the paper. We have tried to incorporate these comments into the revised version of the paper as best as possible. We have uploaded the revised version. Specifically, we added new experiments comparing against CVaR as well as applying the method of Bertsimas and Kallus (2020) to the minimum-violation objective we proposed in this paper. We also present these results on a wider range of quantiles (not only 99\% as we had done in the original version of our paper, but rather quantiles ranging from 85-100\%). We have also performed these experiments for the electricity generation problem. We also analyze how varying $\epsilon$ affects the quality of the decisions, particularly, we examine the trade-off between mean-cost and $q^{th}$ quantile cost.


Moreover, we have also now added  significant discussion about the impact of $\epsilon$ on the bound in
Theorem 4.1 as well as on the learnability of the predictors $\hat{p}_k({x})$. These are important questions raised by the reviewers. We have done our best to address these questions. This discussion can be found following the statement of Theorem 4.1. Having added this, as advised by one of the reviewers, we have also moved Theorem 5.1 from the main text to Section 5 in the appendix.

We would also like to present an alternative viewpoint of our proposed method which connects the choice of objective $\mathbb{E}[\max\{ R_{{\nu}}({w}) - \phi, 0 \}]$ to the rest of the method and how the weights $\hat{p}_{k}({x})$ are generated. This interpretation is unique to our proposed method, and is different from  Bertsimas and Kallus (2020) and related literature. This discussion can be found at the end of Section 3 on page 5 in the revised version of the paper.

---

### Meta-Review · Area_Chair_Qrfe · 2023-12-12

**Metareview:**

This paper introduces a robust optimization framework based on domain discretization concerning the dataset, constructing near-optimal solution sets for each data sample. The subsequent step involves developing a machine learning model to estimate the probability of optimal solutions being near-optimal across various data samples. While demonstrating flexibility in ML model selection, such as k-nearest neighbor, the paper offers theoretical performance guarantees and promising numerical results. To enhance the paper, further exploration of connections with existing data-driven robust optimization methods is suggested, along with a refined presentation of theoretical guarantees and additional insights into the proposed framework for robust optimization problems.

We accept the submission, however, please incorporate the reviewer's comments and clean up the write-ups, correct all the typos, especially in the Appendix.

**Justification For Why Not Higher Score:**

N/A

**Justification For Why Not Lower Score:**

N/A

---

### Decision · Program_Chairs · 2024-01-16

Accept (poster)